# Low $^{13}$C-$^{13}$C abundances in abiotic ethane

Koudai Taguchi [1] ✉, Alexis Gilbert [1,2] ✉, Barbara Sherwood Lollar[3,4], Thomas Giunta[3,5], Christopher J. Boreham[6], Qi Liu [7], Juske Horita[8] & Yuichiro Ueno [1,2,9] ✉

Distinguishing biotic compounds from abiotic ones is important in resource geology, biogeochemistry, and the search for life in the universe. Stable isotopes have traditionally been used to discriminate the origins of organic materials, with particular focus on hydrocarbons. However, despite extensive efforts, unequivocal distinction of abiotic hydrocarbons remains challenging. Recent development of clumped-isotope analysis provides more robust information because it is independent of the stable isotopic composition of the starting material. Here, we report data from a $^{13}$C-$^{13}$C clumped-isotope analysis of ethane and demonstrate that the abiotically-synthesized ethane shows distinctively low $^{13}$C-$^{13}$C abundances compared to thermogenic ethane. A collision frequency model predicts the observed low $^{13}$C-$^{13}$C abundances (anti-clumping) in ethane produced from methyl radical recombination. In contrast, thermogenic ethane presumably exhibits near stochastic $^{13}$C-$^{13}$C distribution inherited from the biological precursor, which undergoes C-C bond cleavage/recombination during metabolism. Further, we find an exceptionally high $^{13}$C-$^{13}$C signature in ethane remaining after microbial oxidation. In summary, the approach distinguishes between thermogenic, microbially altered, and abiotic hydrocarbons. The $^{13}$C-$^{13}$C signature can provide an important step forward for discrimination of the origin of organic molecules on Earth and in extra-terrestrial environments.

Detecting organic molecules synthesised via biological processes and distinguishing them from those synthesised via abiotic processes is critical to the search for life elsewhere in the universe[1–3]. Hydrocarbons have been detected on Mars[4], Enceladus[5], and certain meteorites[6], although their origins remain a matter of debate. On Earth, natural hydrocarbons are mainly biotic in origin, produced either by thermal decomposition of sedimentary organic matter or microbial production including methanogenesis[7,8]. Contrastingly, some abiotic hydrocarbons are produced by a variety of reactions (including free-radical, the Sabatier, and Fischer–Tropsch-type reactions[9–12]) in both deep crustal fluids, hydrothermal systems, and sites of low-temperature water–rock reaction such as serpentinization.

Stable isotopes of carbon and hydrogen have long been used to discriminate hydrocarbon sources[7,8]. Thermogenic and abiotic hydrocarbons can sometimes be distinguished using compound-specific isotopic analysis (CSIA)[10,13]; namely, the relationship between the $^{13}$C/$^{12}$C and $^{2}$H/$^{1}$H ratios of individual hydrocarbons (methane, ethane, propane, and $n$-butane) (Fig. 1b). However, isotopic identification of abiotic hydrocarbons is often challenging (see ref. 11 and references therein), partly because the CSIA requires a set of

[1]Department of Earth and Planetary Sciences, Tokyo Institute of Technology, Meguro, Tokyo 152-8551, Japan. [2]Earth-Life Science Institute (WPI-ELSI), Tokyo Institute of Technology, Meguro, Tokyo 152-8550, Japan. [3]Department of Earth Sciences, University of Toronto, Toronto, ON M5S 3B1, Canada. [4]Institut de physique du globe de Paris (IPGP), Université Paris Cité, Paris, France. [5]Univ Brest, CNRS, Ifremer, Geo-Ocean, F-29280 Plouzané, France. [6]Geoscience Australia, Canberra, ACT, PO Box 378, 2601, Australia. [7]State Key Laboratory of Ore Deposit Geochemistry, Institute of Geochemistry, Chinese Academy of Sciences, Guiyang 550081, China. [8]Department of Geosciences, Texas Tech University, Lubbock, TX 79409, USA. [9]Institute for Extra-cutting-edge Science and Technology Avant-garde Research (X-star), Japan Agency for Marine-Earth Science and Technology (JAMSTEC), Natsushima-cho, Yokosuka 237-0061, Japan. ✉e-mail: taguchi.k.ab@m.titech.ac.jp; gilbert.a.aa@m.titech.ac.jp; ueno.y.ac@m.titech.ac.jp

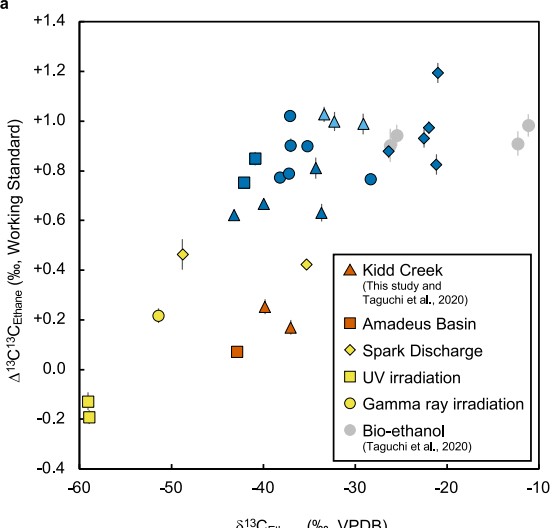

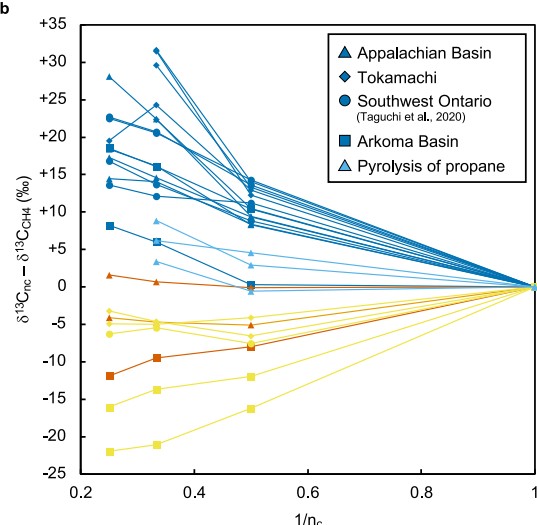

**Fig. 1 | Results of $\Delta^{13}C^{13}C$ and compound-specific isotopic analysis (CSIA) of hydrocarbons ($\delta^{13}C$). a** Relationship between $\Delta^{13}C^{13}C_{Ethane}$ and $\delta^{13}C_{Ethane}$ value normalised against Vienna Pee Dee Belemnite (VPDB). Symbols are the same as shown in the legend of panel **b**. Grey circles denote bio-ethanol from three plants: the C3-type, the C4-type, and the Crassulacean Acid Metabolism (CAM)[19]. Light blue triangles represent ethene ($C_2H_4$) produced by propane pyrolysis (see Methods).

Blue symbols represent proposed thermogenic gases, and orange represent proposed abiotic gases. The error bars are the standard error of the mean. The analytical uncertainty for the $\delta^{13}C$ value is within the symbol. **b** Inverse of carbon number ($n_c$) of individual hydrocarbon versus its carbon isotope composition relative to methane ($\delta^{13}C_{CH4}$). The analytical uncertainty for the $\delta^{13}C$ value is within the symbol.

molecules, all of which are not always available to sample. The recent development of the clumped-isotope analysis allows the collation of information preserved within a single molecule without a need for analysing other related molecules[14–16]. For example, the abundance of clumped isotopes of methane ($^{13}CH_3D$ and $CH_2D_2$) is now routinely used as a geothermometer[17], although the C-H bonding is susceptible to isotopic exchange which can lead, in some cases, to the reset of $^{13}CH_3D$ and $CH_2D_2$ abundances[18]. More robust information may come from $^{13}C$-$^{13}C$ clumping in organic molecules, because carbon in ethane is less readily exchanged than hydrogen, which is exchanged with surrounding water[18].

We have developed a method to determine the relative abundances of the three isotopologues in ethane ($^{12}CH_3^{12}CH_3$, $^{12}CH_3^{13}CH_3$, and $^{13}CH_3^{13}CH_3$) reported as $\Delta^{13}C^{13}C$ values[19,20] (see Methods). Here, we show that this tool can be used to distinguish between abiotic and biotic hydrocarbons. We have examined natural gas ethane from various geological settings and compared them with abiotic ethane synthesised from methane in the laboratory. Based on experimental results and observations, we present a mechanistic understanding of $^{13}C$-$^{13}C$ abundances in hydrocarbons to account for the results found on abiotic and thermogenic ethane.

## Results and discussion
### General trends
Results of our clumped-isotope analysis show that thermogenic natural gas exhibits relatively high $\Delta^{13}C^{13}C$ values (Fig. 1a and Supplementary Table 1) and have a typical $\delta^{13}C$ distribution pattern classically observed for thermogenic gas[13,21], in which each longer-chain alkane is more enriched in $^{13}C$ than the previous alkane (Fig. 1b and Supplementary Table 2). In contrast, abiotic ethane synthesised from $CH_4$ exhibit distinctively low $\Delta^{13}C^{13}C$ values irrespective of the energy source used (i.e., ultraviolet [UV] light, spark discharge, and gamma-ray irradiation: see Methods). The low $\Delta^{13}C^{13}C$ values are also seen in hydrocarbons from deep fracture fluids in Kidd Creek (Canada) and the Dingo gas field in the Amadeus Basin (Australia) (see Methods for geologic settings), also proposed to have an abiotic origin.

A previous study using high-resolution mass spectrometry exhibited 4‰ variation of $\Delta^{13}C^{13}C$ values of thermogenic ethane[22]. This

variation has been suggested to arise from the pyrolysis of ethane, which leads to a decrease in $\Delta^{13}C^{13}C$ values[22,23]. The present study using a conventional isotope ratio mass spectrometry after conversion of $C_2H_6$ to $C_2F_6$ shows a narrower range of the $\Delta^{13}C^{13}C$ values (0.57‰). Our pyrolysis experiment conducted at the same temperature as in ref. 22 (600 °C) and using a similar quartz vessel showed no change in the $\Delta^{13}C^{13}C$ values in contrast to ref. 22 (Supplementary Fig. 1 and Supplementary Table 3). These observations point to a potential discrepancy between the two methods for isotopologues analysis. Further interlaboratory comparisons will be necessary to calibrate the data from the two methods. The data presented here will be obtained solely by the method presented in ref. 19 that gives reproducible $\Delta^{13}C^{13}C$ values with no scale compression[20] (see Methods).

### Thermogenic ethane
The $\Delta^{13}C^{13}C$ value of thermogenic ethane could be attributed to C-C bonding in precursor molecules and the kinetic isotope effect during thermal cracking, as discussed in previous studies[22,23] (Fig. 2a). In an ideal case where ethane is produced by breaking at least one C-C bond in an organic precursor, the kinetic isotope effect is relevant only to one carbon in ethane resulting in preferential $^{13}C$ enrichment in one of the two carbons. In this case, the intramolecular bias in $^{13}C$ lowers the $\Delta^{13}C^{13}C$ value owing to the combinatorial effect[24,25], even though the two carbons in ethane are symmetrically equivalent. Combinatorial isotope effects are statistical clumped-isotope anomalies that occur when two atoms at two different positions in a single molecule isotopically differ[24,25]. In other words, when $^{13}C$ is not evenly distributed at the two positions in an ethane molecule, the abundances of $^{13}C$-$^{13}C$ are lower than the stochastic (random) distribution. This does not apply to molecules with non-equivalent atomic sites, typically ethanol, for which an accurate stochastic distribution can be calculated based on the $^{13}C/^{12}C$ ratio of both sites.

We estimated the intramolecular bias of thermogenic ethane based on a simple model in which hydrocarbons are formed by the cleavage of at least one C-C bond in an $n$-alkyl chain, as the simplest case considered in previous model[21–23]. For ethane, the $\delta^{13}C$ (=[($^{13}C/^{12}C)_{sample}/(^{13}C/^{12}C)_{standard}$] – 1) of one C-atom ($C_1$) is that of the original precursor, whereas that of the other C-atom ($C_2$) is altered by the

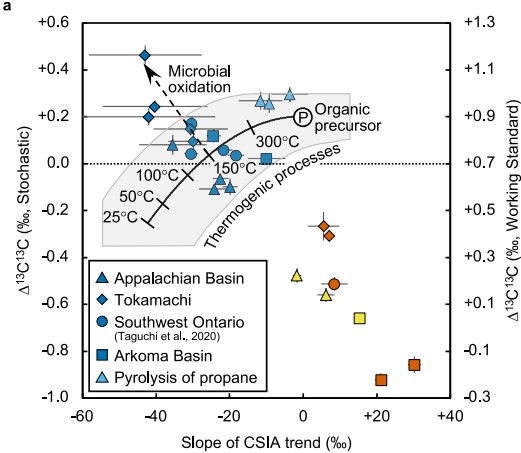

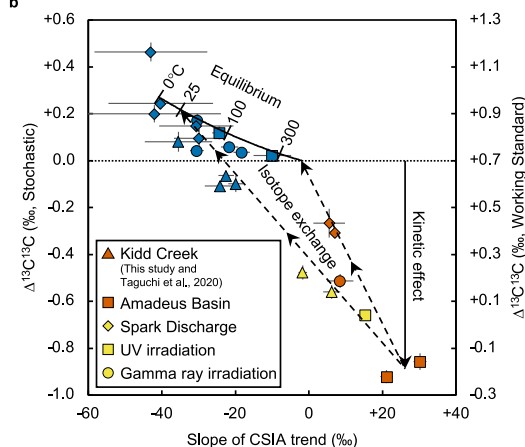

**Fig. 2 | Δ¹³C¹³C vs. slope of compound-specific isotopic analysis (CSIA) trend.** The horizontal axis shows the slope of the relationship between $\delta^{13}C$ and $1/n_c$ (derived from Fig. 1b), which also assumed to be an intrinsic $\delta^{13}C$ bias between the two positions of carbon in a molecule (see text). The $\Delta^{13}C^{13}C$ scale to the stochastic distribution was estimated by assuming that the C-C bonds of biological glucose are under homogeneous isotopic equilibrium (see Methods). The error bars are the standard error of the mean. **a** The curved black line shows the predicted $\Delta^{13}C^{13}C$ value of ethane produced by C-C bond cleavage from an organic precursor (denoted as 'P') at each temperature, considering the combinatorial effect (see Methods). The CSIA slope at each temperature is also calculated using the same thermal cracking model[21–23,26] (see Methods). The grey shaded area shows the uncertainty of the calculation, mainly derived from the possible range in $\Delta^{13}C^{13}C$ and differences in $\delta^{13}C$ values between two adjacent positions of precursor molecules for the thermogenic hydrocarbons (see Methods). The dotted arrow shows the expected change due to microbial oxidation of ethane for Tokamachi mud volcano (Supplementary Fig. 2). **b** The curved black line shows the theoretically calculated $\Delta^{13}C^{13}C$ value of ethane at each temperature (Supplementary Fig. 3 and Supplementary Table 5) and $\delta^{13}C$ values of hydrocarbons[67,68] (see Methods). The dotted arrows show the notional changes due to isotope exchange among hydrocarbons after formation.

cracking process. This creates an intramolecular bias within the ethane molecule, denoted as $\Delta^{13}C_{Ethane}$ ($= \delta^{13}C_{C1} - \delta^{13}C_{C2}$). The $\Delta^{13}C_{Ethane}$ value can be obtained through the slope of the relationship between the $\delta^{13}C$ values of individual hydrocarbons and the inverse of their carbon number (i.e., $1/n_c$) (see Methods) (Fig. 1b)[21–23]. The combinatorial effect ($\Delta^{13}C^{13}C_{Comb}$) can therefore be calculated as follows (see Methods for detailed calculation):

$$\Delta^{13}C^{13}C_{Comb} = -\left(\Delta^{13}C_{Ethane}/2(\delta^{13}C_{Ethane}+1000)\right)^2 \times 1000, \quad (1)$$

where $\delta^{13}C_{Ethane}$ represents the carbon isotope composition of ethane. Assuming a typical $\delta^{13}C_{Ethane}$ value of −40‰ and a kinetic isotope effect ($^{13}k/^{12}k$) from 0.958 to 0.986 (i.e., $\delta^{13}C_{C1} > \delta^{13}C_{C2}$), corresponding to temperatures ranging from 25 to 300 °C[26], the $\Delta^{13}C^{13}C_{Comb}$ is predicted to range from −0.04 to −0.45‰ (Fig. 2a). Assuming a cracking temperature of 100 °C, the $\Delta^{13}C^{13}C$ value of ethane would be lower than −0.25‰ relative to the $\Delta^{13}C^{13}C$ value of precursor hydrocarbons. Estimating the $\Delta^{13}C^{13}C$ value of the original precursors is difficult, since measurement of $^{13}C$-$^{13}C$ clumped isotopes has not been applied to natural gas precursors, such as alkanes and kerogen. The only available data for biological molecules to date are derived from bio-ethanol (Fig. 1a)[19]. Despite the different types of photosynthetic pathway (the C3-type, the C4-type, and the Crassulacean Acid Metabolism [CAM]), the bio-ethanol show a narrow range of $\Delta^{13}C^{13}C$ values from +0.90 to +0.98‰, suggesting bio-ethanol is a good representative of biological molecules[19]. Using the $\Delta^{13}C^{13}C$ value of the ethanol as a starting estimate for an organic precursor (denoted as 'P' in Fig. 2a), the $\Delta^{13}C^{13}C$ value of thermogenic ethane can be calculated (Fig. 2a). Remarkably, the observed $\Delta^{13}C^{13}C$ values of the majority of the thermogenic natural gas ethanes in this study fall within the range predicted by this model at about 150 °C (Fig. 2a). The thermogenic ethane seems aligned perpendicular to the equilibrium temperature curve (Fig. 2b), which may potentially reflect the variation of $\Delta^{13}C^{13}C$ of the organic precursor, though, at present, the available $\Delta^{13}C^{13}C$ data of the organic precursor is only limited to bio-ethanol. Future studies should pursue $\Delta^{13}C^{13}C$ of organic molecules such as n-alkanes, fatty acids, and lignin to evaluate the $\Delta^{13}C^{13}C$ variations in the organic precursor.

## Microbial oxidation of ethane

Three natural gas samples from Tokamachi show $\Delta^{13}C^{13}C$ values exceeding the predicted range of thermogenic ethane from this model (Fig. 2a). The high $\Delta^{13}C^{13}C$ in these samples could be due to microbial oxidation of ethane. Microbial strains capable of metabolising non-methane hydrocarbons (NMHCs) reportedly inhabit various geological settings under anoxic conditions[27–29]. The samples showing the high $\Delta^{13}C^{13}C$ values also have high $C_1/(C_2 + C_3)$ ratios, consistent with the anaerobic oxidation of NMHCs (Supplementary Fig. 2)[30]. Furthermore, at each of these localities, the high $\Delta^{13}C^{13}C$ in ethane correlated with the $^{13}C$ enrichment of the central carbon in propane, strongly indicating microbial degradation of NMHCs (Supplementary Fig. 2 and Supplementary Table 4)[31]. Microbial oxidation of NMHCs is known to yield increased $^{13}C$ enrichment in higher hydrocarbons[31] (i.e., the steeper slope of the CSIA trend seen in Fig. 1b), consistent with the higher $\Delta^{13}C^{13}C$ values of ethane indicative of anaerobic oxidation origin.

Although the $\Delta^{13}C^{13}C$ analysis of pure culture anaerobic oxidation remains unreported, a conjecture can nevertheless be made based on recent findings on the anaerobic oxidation of ethane by archaea[28,29]. Previous culture experiments on ethane-oxidising archaea have shown that the enzymatic steps from ethane to $CO_2$ by the archaea 'Candidatus Ethanoperedens' can be reversible within the cell[29]. In such case, a reverse reaction in which some $CO_2$ is converted back into ethane may yield isotopic bond re-ordering towards thermodynamic equilibrium. Note that a similar mechanism has been proposed for the anaerobic oxidation of methane and is believed to promote isotopic re-equilibration in the residual methane[32]. Accordingly, it is conceivable that residual ethane undergoing microbial oxidation would approach isotopic equilibrium ($\Delta^{13}C^{13}C \geq +0.10‰$ relative to the stochastic distribution of ethane; Fig. 2b, Supplementary Fig. 3 and Supplementary Table 5) at the microbiologically functional temperature range (<122 °C[33]) based on the assumption of stochastic reference frame (see Methods). A more precise estimate of the $\Delta^{13}C^{13}C$ change in microbial degradation is not possible at this stage, owing to the uncertainty in the growth temperature and degree of reversibility in natural populations. Nonetheless, the observed $\Delta^{13}C^{13}C$ increase seen in the three samples here is consistent with microbial degradation of

ethane (Fig. 2b). Future studies should pursue $\Delta^{13}C^{13}C$ analysis to evaluate microbial activity pertaining to the anaerobic oxidation of ethane. Because the anaerobic oxidation of ethane increases $\Delta^{13}C^{13}C$ and its direction of the trend is away from those of abiotic ethane, the ability of $\Delta^{13}C^{13}C$ to distinguish between abiotic and thermogenic ethane is not impaired. Other processes potentially alter the $\Delta^{13}C^{13}C$ value in ethane and are compatible with the observed variation of thermogenic ethane. These include diffusion (an increase of $\Delta^{13}C^{13}C$ value by 0.3‰ in the case where molecular collision is important), mixing with different sources (an increase of $\Delta^{13}C^{13}C$ value by up to 0.13‰ in the case of mixing samples with the same $\Delta^{13}C^{13}C$ values but with different $\delta^{13}C$ values of −20‰ and −45‰) and secondary cracking of ethane itself (no $\Delta^{13}C^{13}C$ variations at 600 °C in this study; see Supplementary Fig. 1)[22,23]. However, again, the discrimination potential of the $\Delta^{13}C^{13}C$ value of ethane is not weakened, because all these factors tend to increase the $\Delta^{13}C^{13}C$ value.

## Abiotic hydrocarbon synthesis

In the stochastic reference frame assumed here (see Methods), the observed negative $\Delta^{13}C^{13}C$ values (i.e., anti-clumping[25]) of abiotic ethane can be explained by kinetic isotope effects governed by collision frequency between $CH_3$ radicals (Fig. 2b). A radical–radical combination such as $CH_3 + CH_3$ is typically a barrierless reaction[34], associated with mild kinetic isotope effects, depending on the ratio of collision frequencies[35]. These collision frequencies are scaled exactly by the inverse square root of the reduced mass ($\mu$) of the collision pair ($m_1$ and $m_2$, respectively), i.e., $\mu = 1/m_1 + 1/m_2$. Thus, the kinetic isotope effect due to the difference in collision frequency can be obtained from the relative reaction rate, calculated from the reduced mass ratio. In the case of a collision between $CH_3$ radicals, the kinetic isotope effects are calculated to be $k'/k = 0.9842$ and $k''/k = 0.9682$, where the prime and double prime refers to singly and doubly substituted isotopologues, respectively. If kinetic isotope effects follow a stochastic distribution, $(k''/k)/(k'/k)^2$ must be equal to 1 (see Supplementary Note 1). Any deviation from unity leads to $\Delta^{13}C^{13}C$ values different from 0. The intrinsic kinetic clumped-isotope effect calculated from the reduced mass of methyl radicals corresponds to a $\Delta^{13}C^{13}C$ value of −0.52‰. A more accurate quantitative estimate requires calculations that include a configurational effect, as isotopic substitutions can affect the minimum energy configuration of the transition state by changing its centre of mass[36]. However, the simple collision frequency calculation shows that $\Delta^{13}C^{13}C$ should be negative (anti-clumping) compared to a stochastic distribution during the methyl radical recombination reaction. Notably, a combinatorial isotope effect is not expected in the case of a methyl radical recombination since the two methyl radicals arise from the same reservoir. However, if one considers surface reactions such as Fischer–Tropsch-type reactions, the mechanism itself may lead to an intramolecular bias in ethane[37], resulting in even lower $\Delta^{13}C^{13}C$ values. The data presented here provide, for the first time, a strong indicator of abiogenecity for ethane, based on the observed pattern of $^{13}C$-$^{13}C$ anti-clumping.

## Isotope exchange in abiotic synthesis

After hydrocarbons are produced, their destruction enhances carbon exchange among individual hydrocarbons, which may potentially lead partial isotopic equilibrium through repeated production and destruction cycling. In our UV irradiation experiments of methane, $C_2H_6$ was produced from the combination of $CH_3$ radicals ($CH_3$ being generated through the reaction of $CH_4$ with OH radicals derived from the photodissociation of water[38]) by the following reaction[39]:

$$CH_3 + CH_3 + M \rightarrow C_2H_6 + M \qquad (2)$$

where M represents any third-body collision partner. In the wavelength range of UV light used in this study, neither $CH_4$, $C_2H_6$, nor $C_3H_8$

photodissociates[38]. In addition, the C-C bonds in $C_2H_6$ and $C_3H_8$ are not decomposed by reactions with OH, H, O, and $CH_3$ radicals, all of which should be present during UV experiments[38]. Hence, the $C_2H_6$ and $C_3H_8$ produced by UV irradiation of methane are unlikely to undergo C-C bond decomposition because of the lack of high-energy photon below 150 nm in our experimental setting. Conversely, for spark discharge and gamma-rays experiment, the C-C bonds of $C_{2+}$ hydrocarbons frequently cleave after their formation as follows[40] (Supplementary Table 6):

$$C_2H_6 \rightarrow 2CH_3 \qquad (3)$$

$$C_3H_8 \rightarrow CH_3 + C_2H_5 \qquad (4)$$

$$C_3H_8 \rightarrow CH_4 + C_2H_4 \qquad (5)$$

In the case of gamma-ray irradiation of methane, the calculated dosage was sufficient to decompose the $C_{2+}$ hydrocarbons formed from methane, although $C_2H_4$ was not detected[12]. Ethane in the spark discharge and gamma-ray irradiation experiments is produced not only by the $CH_3$ radical polymerisation but also by the $C_{3+}$ hydrocarbon decomposition. After production, ethane decomposes to $CH_3$, implying that the $C_1$ polymerisation is not unidirectional. The C-C chain elongation and shortening may alter the $\Delta^{13}C^{13}C$ values of ethane to an extent depending on the degree of reversibility. Fully reversible reactions may yield equilibrium isotope composition (the curved black line in Fig. 2b), whereas irreversible reactions tend to be governed by kinetic isotope effect as represented in the ethane synthesised by UV experiment (Fig. 2b). We suggest that cleavage of C-C bonds in hydrocarbons may enhance the reversibility and leads to an isotopic exchange, where $\Delta^{13}C^{13}C$ of abiotic ethane shifts toward the homogeneous isotopic equilibrium ($\Delta^{13}C^{13}C = +0.22‰$ at 25 °C; Supplementary Fig. 3) (Fig. 2b).

## $\Delta^{13}C^{13}C$ systematics of abiotic hydrocarbons

In summary, the observed low $\Delta^{13}C^{13}C$ values in abiotic ethane can be explained by anti-clumping due to the kinetic isotope effect at the C-C bond formation (negative $\Delta^{13}C^{13}C$). Subsequent isotope exchange facilitated through the backreaction from the higher hydrocarbons during the polymerisation sequence may cause the increase in $\Delta^{13}C^{13}C$ value (Fig. 2b). The two processes likely occur naturally and seem applicable to the abiotic hydrocarbons from Kidd Creek fracture fluids and the Dingo gas field (see Methods), both of which exhibit low $\Delta^{13}C^{13}C$ values within the expected abiotic range (Fig. 2b). The similarity of ethane from the gamma radiolysis experiments to the Kidd Creek samples is notable given the proposed role of radiolysis in producing acetate and formate at that site[41]. If isotope exchange continues, leading to homogeneous isotopic equilibrium, the $\Delta^{13}C^{13}C$ of abiotic ethane may eventually become indistinguishable from that of thermogenic hydrocarbons. However, the $^{13}C$-$^{13}C$ anti-clumping observed in the natural gases from the two sites (Kidd Creek and the Dingo gas field) demonstrates that the distinctively low abiotic $\Delta^{13}C^{13}C$ signature survives in nature. Based on these findings, $^{13}C$-$^{13}C$ anti-clumping in ethane can be a valuable approach to distinguish abiotic hydrocarbons from thermogenic, and potentially from microbial sources. The $^{13}C$-$^{13}C$ signature may thus be applied in investigations of the origin of ethane in terrestrial and extra-terrestrial settings. Moreover, not only ethane but a variety of organic molecules containing C-C bonds can be subjected to this analytical approach to distinguish abiotic formation pathways, in geological and even extra-terrestrial settings, such as Mars, Titan, and Enceladaus[2,5,42].

## Methods

### $\Delta^{13}C^{13}C$ notation

The abundance of $^{13}C$-$^{13}C$ isotopologues was conventionally reported as a deviation from the stochastic abundance of the isotopologues:

$$\Delta^{13}C^{13}C \equiv {}^{1313}R_{sample}/{}^{1313}R_{stochastic} - 1 \qquad (6)$$

where $^{1313}R$ is defined as the abundance of $^{13}C$-$^{13}C$ isotopologues compared to $^{12}C$-$^{12}C$ isotopologues, and $R_{stochastic}$ refers to the abundance ratio in a random distribution. The stochastic distribution of $C_2$ isotopologues is calculated as follows:

$$^{1313}R_{stochastic} = {}^{13}R \times {}^{13}R \qquad (7)$$

where $^{13}R$ indicates the $^{13}C/^{12}C$ ratio in all $C_2$ molecules. In this study, we report $\Delta^{13}C^{13}C'$ as the natural logarithm of $\alpha$:

$$\Delta^{13}C^{13}C' \equiv \ln(\alpha) \approx (\alpha - 1); (\text{since } \alpha \approx 1) \qquad (8)$$

where $\alpha$ represents the equilibrium constant of the homogeneous isotope exchange reaction:

$$2^{12}C^{13}C \leftrightarrows {}^{12}C^{12}C + {}^{13}C^{13}C \qquad (9)$$

$$\alpha \equiv [^{12}C^{12}C][^{13}C^{13}C]/[^{12}C^{13}C]^2 \approx {}^{1313}R/(2 \times {}^{13}R)^2 \qquad (10)$$

where $^{13}R$ is calculated from the isotopologue ratio of $[^{12}C^{13}C]/[^{12}C^{12}C]$ divided by 2, reflecting the symmetry of two carbon atoms in ethane. The $\Delta^{13}C^{13}C'$ value is approximately equal to that of the conventional $\Delta^{13}C^{13}C$ (Eq. (6)). In the case of $C_2$ compounds, however, it is difficult to determine the $^{1313}R_{stochastic}$ accurately because C-C bond breaking and recombination do not usually occur reversibly. Thus, calibrating the value with experiments at different temperatures is not feasible. Therefore, $\Delta^{13}C^{13}C'^*$ value is calculated as follows:

$$
\begin{aligned}
\Delta^{13}C^{13}C'^* &\equiv \Delta^{13}C^{13}C'_{sample} - \Delta^{13}C^{13}C'_{reference} \\
&= \ln({}^{1313}R_{sample}/{}^{13}R_{sample}^2) - \ln({}^{1313}R_{reference}/{}^{13}R_{reference}^2) \\
&= \ln({}^{1313}R_{sample}/{}^{1313}R_{reference}) - 2 \times \ln({}^{13}R_{sample}/{}^{13}R_{reference}) \\
&= \delta^{13}C^{13}C' - 2 \times \delta^{13}C'
\end{aligned} \qquad (11)
$$

where $\delta^{13}C^{13}C'$ and $\delta^{13}C'$ represent the ratio of $^{1313}R$ and $^{13}R$ among sample and reference gases as the natural logarithm. Note that all isotope values ($\Delta^{13}C^{13}C'^*$, $\delta^{13}C^{13}C'$ and $\delta^{13}C'$) are expressed in ‰.

### Measurement of $\Delta^{13}C^{13}C$

We used a fluorination method for the measurement of $^{13}C$-$^{13}C$ species[19,20], which is based on the fluorination of $C_2$ compounds to hexafluoroethane ($C_2F_6$) and subsequent measurement of its $^{13}C$ isotopologues with a conventional isotope ratio mass spectrometer. The purified $C_2F_6$ was introduced into a mass spectrometer (Thermo Fischer MAT253), used in the conventional dual inlet mode. The typical standard deviation of the mean for $\delta^{13}C'$, $\delta^{13}C^{13}C'$, and $\Delta^{13}C^{13}C'^*$ values was ±0.01‰, ±0.09‰, and ±0.09‰, respectively ($n = 6$).

Consequently, we define the empirical transfer function as follows:

$$\Delta^{13}C^{13}C'^*_{CSC} = \lambda \times \Delta^{13}C^{13}C'^* \qquad (12)$$

where $\Delta^{13}C^{13}C'^*_{CSC}$ value refers to the 'true scale' value, corrected for scale compression. The $\lambda$ value is $1.2541 \pm 0.0101$ for ethanol and ethene, whereas 1 for ethane since the latter is not prone to scrambling which may have occurred during the fluorination of ethene but not in the ion source of the mass spectrometer[20]. For simplicity, we will describe the corrected $\Delta^{13}C^{13}C'^*_{CSC}$ as $\Delta^{13}C^{13}C$ in the following and main text.

### Reference frame for $\Delta^{13}C^{13}C$ measurement

The $\Delta^{13}C^{13}C$ values obtained as described above are relative and are not referred to against the stochastic distribution but a working standard ($C_2F_6$)[19,20]. To obtain values reported against a stochastic distribution, we constructed a reference frame for $^{13}C$-$^{13}C$ isotopologues analysis by estimating that C-C bonds in biological glucose could be under thermodynamic isotope equilibrium. The $\Delta^{13}C^{13}C$ of ethanol produced by the fermentation of sugars from different plants ($C_3$, $C_4$, and CAM) is reportedly uniform, whereas the $^{13}C/^{12}C$ ratio and position-specific isotope composition vary[19]. The latter has been explained by differences in $CO_2$ assimilation[43] and internal metabolic fluxes[44]; the narrow range of $\Delta^{13}C^{13}C$ values can be explained by the metabolic origin of sugars, i.e., the Calvin–Benson–Bassham (CBB) cycle[19]. The C-C bonds of glucose are derived from the carboxylation, aldolisation, and transketolisation reactions in the CBB cycle. Enzymes facilitate these reactions, and while it is known that the carboxylation reaction is irreversible, the aldolisation and ketolisation reactions are at equilibrium[45]. The measured ethanol[19] is derived from the carbons in the $C_1$-$C_2$ and $C_5$-$C_6$ positions of glucose which are produced in an aldolisation reaction[45]. In this scenario, the $\Delta^{13}C^{13}C$ values of ethanol could be controlled by thermodynamic equilibrium, which enriches the doubly substituted isotopologue, leading to a positive $\Delta^{13}C^{13}C$ value compared to the stochastic distribution.

Because glucose is a complex molecule, we performed theoretical calculations to estimate equilibrium distributions of isotopologues of $CH_2$=$CH_2$, $CH_3$-$CH_3$, $CH_3$-$CH_2OH$, and $CH_2OH$-$CH_2OH$ as model molecules with C-C bonds (Supplementary Fig. 3 and Supplementary Table 5). The clumped-isotope signature at an isotopic equilibrium was calculated by applying the Bigeleisen–Mayer equation[46,47]. The molecular constants within the Bigeleisen–Mayer equation were obtained through quantum chemical calculations. We used B3LYP/6-311+G (d, p) level[48–50] for geometry optimisations, single-point energy calculations, and harmonic frequency generations. All calculations were performed using the Gaussian 09 package[51]. 'Very tight' geometry convergence criteria and 'superfine' grids built in Gaussian 09 were applied for geometry optimisation procedures and further computations.

The calculated $\Delta^{13}C^{13}C$ values for ethane were in good agreement with the previous studies[52]. The obtained $\Delta^{13}C^{13}C$ differed by up to 0.02‰ among $CH_3$-$CH_3$, $CH_3$-$CH_2OH$, and $CH_2OH$-$CH_2OH$ molecules; this difference is sufficiently low compared to the analytical accuracy of $\Delta^{13}C^{13}C$ (Supplementary Fig. 3 and Supplementary Table 5). Thus, we used data obtained from $CH_2OH$-$CH_2OH$ to predict $\Delta^{13}C^{13}C$ of the $C_1$-$C_2$ and $C_5$-$C_6$ bonds in glucose under thermodynamic equilibrium, because the carbons in the $C_1$-$C_2$ and $C_5$-$C_6$ positions of glucose are composed as $CH_2OH$-CHOH- and CHO-CHOH-, respectively. The $\Delta^{13}C^{13}C$ of the used $C_2F_6$ standard gas in this study was −0.72‰ compared to the stochastic distribution by assuming that the average of $\Delta^{13}C^{13}C$ of bio-ethanol (+0.93‰) is in thermodynamic equilibrium and using the results of the theoretical calculations for $CH_2OH$-$CH_2OH$ (+0.2‰ at 25 °C). In natural ethanol, the $\delta^{13}C$ values in positions $CH_3$ and $CH_2OH$ can be different by up to 11.4‰, which would result in lower $\Delta^{13}C^{13}C$ values compared with the stochastic distribution owing to combinatorial effects[25]. However, the combinatorial effect in ethanol measured in this study is up to −0.03‰ considering a site-specific $^{13}C$ distribution in ethanol of 11.4‰ at maximum[53]. The combinatorial effects calculated here are much lower than the analytical precision of ±0.09‰ and can thus be quantitatively neglected here.

### Measurement of $\delta^{13}C$ values of hydrocarbons

The $\delta^{13}C$ values were determined using gas chromatography coupled with isotope ratio mass spectrometry (DeltaplusXP, Thermo Fisher Scientific K.K., Tokyo, Japan) via a combustion furnace and a conflow

interface (GC Combustion III, Thermo Fisher Scientific K.K., Tokyo, Japan) (GC-C-IRMS). The gas chromatography column used was HP-PLOT-Q (30 m × 0.32 mm i.d., 10 μm film thickness; GL Sciences Inc., Tokyo, Japan), and the carrier gas was high-purity helium (99.999%; Fujii Co.). The conditions of the GC oven were as follows: injector temperature 250 °C; split mode (variable split ratio); flow rate 1.5 mL/min; oven temperature programme 50 °C (maintained for 5 min) raised to 200 °C (maintained for 10 min) at a rate of 10 °C/min. The combustion furnace consisted of a ceramic tube packed with CuO, NiO, and Pt wires, operating at 960 °C. Isotopic standardisation was made by $CO_2$ injections calibrated against the natural gas standard NGS-2 provided by the National Institute of Standards and Technology (NIST), Gaithersburg, MD, USA.

## Measurement of intramolecular $\delta^{13}C$ composition in propane
Intramolecular $\delta^{13}C$ bias of propane was determined by an online pyrolysis system coupled with GC-C-IRMS[54]. Isotopic standardisation was made by $CO_2$ injections calibrated against the NIST natural gas standard NGS-2. The relative $^{13}C$ enrichment in a given position is defined as the difference in isotopic composition between central and terminal carbon positions. Three fragments are used for its calculation: $CH_4$, $C_2H_4$, and $C_2H_6$. $CH_4$ and $C_2H_6$ arise from the terminal position only, while $C_2H_4$ arises from an equal contribution of terminal and central positions. The relative $^{13}C$ enrichment in the central position (=$\Delta^{13}C_{propane}$, expressed in ‰) is defined as follows:

$$\Delta^{13}C_{propane} = \delta^{13}C_{central} - \delta^{13}C_{terminal} \quad (13)$$

where $\delta^{13}C_{central}$ and $\delta^{13}C_{terminal}$ refer to the carbon isotope composition of central and terminal positions, respectively.

## Purification of ethane and ethene
Ethene and ethane must be purified before the fluorination reaction[19,20]. These gases were purified using a gas chromatograph GC-4000 plus (GL Sciences Inc., Tokyo, Japan) equipped with Hayesep Q column (1/8" od., 60/80 mesh, 4 m; GL Sciences Inc., Tokyo, Japan) connected to a vacuum line. The sample was directly introduced into the system using a gas-tight syringe through a rubber septum. In both cases, condensable products were trapped at –196 °C (liquid nitrogen), and the remaining non-condensable gases were evacuated under a vacuum. The condensed products were then released with a water bath at room temperature (25 °C) before being introduced into the gas chromatograph. High-purity helium (99.999%; Fujii Co.) was used as the carrier gas. Ethane and ethene could be separated and collected through the 6-port switching valve due to their different retention times. Other impurities were discarded. The conditions of the GC oven were as follows: injector temperature 120 °C; column pressure1 200 kPa column; pressure2 160 kPa at 80 °C; oven temperature programme 35 °C (maintained for 15 min) raised to 200 °C (maintained for 16 min) at a rate of 60 °C/min.

## UV irradiation experiment
UV irradiation of methane was conducted in a glass flask (457 mL) used in a previous study[38]. The top of the flask is made of UV-grade synthetic quartz window, which is transparent for >175 nm photon. A high-pressure xenon arc lamp (Xe lamp: Cermax, CX-04E, output setting 20 A) was used as the UV source, with a solar-like UV spectrum used in a previous report[38]. Before UV irradiation, 50 mL of doubly distilled water was injected through a syringe port. The water was frozen using liquid nitrogen, and the remaining gas was evacuated from the vacuum line to remove the $CO_2$ or $O_2$ trapped in the water. Then, methane (purity 99.9%, GL Science Inc.) was introduced without purification into the flask from the vacuum line at 25 °C to a pressure of about 12 kPa. After introducing methane gas, the flask was kept at 25 °C using a water bath (MC-1, ASONE). An aliquot of gas phase was sampled from

the vacuum line to measure the chemical and carbon isotope composition before the UV irradiation (0 h). In this experiment, UV light was irradiated vertically from the top to the liquid water surface. Methane was exposed under UV light for 3 and 16 h. After the irradiation, the gas sample was collected from the vacuum line to a stainless-steel finger.

## Spark discharge experiment
$C_{2+}$ hydrocarbons were produced by spark discharge of methane gas. This experiment was conducted in a glass flask (457 mL) previously evacuated through a vacuum line. Methane (purity 99.9%, GL Science Inc.) was introduced without purification into the flask using a gas-tight syringe to a pressure of 12 kPa. The glass flask was connected with a tungsten pole through Swagelok, and the pole was connected with a spark discharger (BD-50E Heavy Duty Generator). Spark discharge of methane was conducted cyclically for 15 min and then stopped for 15 min to avoid elevating the temperature of the reaction vessel. The two experiments were conducted at room temperature (25 °C) with a total duration of 15 min and 5 h. Output adjustment of spark discharger was controlled at eight levels. After exposing methane under spark discharge, gas samples were collected through the vacuum line to a stainless-steel finger.

## Gamma-ray irradiation experiment
The gamma-ray irradiation experiment was undertaken at ANSTO, Australia[12,55], where $C_2$-$C_5$ light hydrocarbons were synthesised via the $^{60}Co$ gamma-ray radiolysis of methane. The sample used in this study was irradiated for 650 h, and the temperature was maintained at 21 °C.

## Ethane pyrolysis
Ethane (purity 99.9%, GL Science Inc.) was decomposed by heating at 600 °C in a muffle furnace. Ethane was introduced into quartz tubes through a vacuum line using liquid nitrogen. Once sealed with a gas burner supplied with oxygen, the tubes were heated at 600 °C in a muffle furnace. The remaining ethane in the tubes was isolated from the other reaction products and measured by manometry to calculate the percentage of gas remaining and measured for its carbon isotope and clumped-isotope composition (Supplementary Fig. 1 and Supplementary Table 3).

## Pyrolysis of propane
Propane pyrolysis was conducted to produce hydrocarbons. Propane gas (purity 99.9%, GL Science Inc.) was introduced into empty Pyrex tubes through a vacuum line using liquid nitrogen. Once sealed with a gas burner, the tubes were heated at 500 °C in a muffle furnace. The chemical and isotopic compositions of the reaction products were characterised and measured using the methods presented above (Supplementary Table 7). Then, the produced ethene in the tubes was purified from the other reaction products to measure the $\Delta^{13}C^{13}C$. Ethene was measured instead of ethane because the latter potentially arises from the recombination of $CH_3$ fragments (2 $CH_3 \rightarrow C_2H_6$), not directly from propane cracking, contrary to ethene[54].

## Natural gas samples
We analysed samples from different natural gas reservoirs: Southwest Ontario Basin (Canada)[31], Appalachian Basin (United States)[56,57], and Arkoma Basin (United States)[58]. Based on isotope composition, hydrocarbons in these basins are suggested to be mainly of thermogenic origin (Fig. 1b). We also analysed gases from the Kidd Creek and the Dingo gas field (Amadeus Basin). CSIA suggests that these hydrocarbons are of abiotic origin (Fig. 1b). In addition, we collected natural gas samples from the Tokamachi mud volcano, which is situated in the Tertiary sedimentary basin in Niigata Prefecture, Central Japan. Niigata Basin is part of the wider Green Tuff belt of Honshu, one of the most important petroleum (oil and gas) producing areas in Japan[59].

**Southwest Ontario Basin**. The sedimentary strata of Southwestern Ontario consist of Late Cambrian to Devonian sediments. The samples analysed in this study are of Silurian to Middle Ordovician age and were collected between June 2012 and December 2013. For more details on the basin and sampling method, please refer to a previous study[31].

**Appalachian Basin**. Samples from the Appalachian Basin are gas seeps collected south of Lake Erie in upstate New York in June 2018. In this region, natural gas seeps are abundant and generally found bubbling in small water ponds or riverbeds on fractured shales of Ordovician and Devonian ages[56,57]. Specifically, samples collected and measured here are those defined in ref. 56 as: Amherst State Park, Barcelona Gas Spring, Chestnut Ridge Eternal Flame, Gasport, and from Pipe Creek. These samples all have signatures of thermogenic generation with methane $\delta^{13}C$ ranging between −42 and −52‰, low $C_1/C_{2+}$ ranging from 1.5 to 10, and ethane concentrations of 7–24 vol.%. The samples were collected by setting on the gas seeps an inverted funnel connected to Tygon tubing and a flow-through gas chamber to avoid air contamination. Gases are then sampled from the gas chamber with a gas-tight syringe and added to a pre-evacuated 60 cc serum vial pre-poisoned with $HgCl_2$.

**Arkoma Basin, Oklahoma, USA**. Hydrocarbons, the main material for shale gas, have been developed from late Devonian and early Mississippian formations, such as the Woodford Shale in Oklahoma and the Chattanooga Shale in Arkansas. Gas was collected from gas wells using standard well-sampling techniques[58].

**Tokamachi mud volcano, Niigata, Japan**. Tokamachi area consists of two active mud volcanoes, Murono and Gamo, located 10 km west of Tokamachi village. At Murono, groundwater, mud, and gases erupt at the bubbling crater; natural gas also seeps from cracks formed in the asphalt pavement along the road of a car test track built around the mud volcano; the crack-seepage mainly occurs in at least two sites. At Gamo, only two small mud craters were found during the survey performed in this study. The gas samples were sampled using the water-displacement method and stored in a glass vial sealed with a butyl rubber septum.

**Kidd Creek, Timmins, Ontario, Canada**. Sampling and characterisation of fracture fluids located at 2.4 km below the surface in a mine operating to 3 km depth in the 2.7-billion-year-old rocks of the Canadian Shield in a Cu-Ag-Zn deposit (stratiform volcanogenic massive sulfide) hosted in interlayered felsic, mafic, ultramafic and metasedimentary deposits that form part of Abitibi greenstone belt as described in previous studies[10,41,60–64]. We analysed ethane samples from a borehole at 7850′ level in 2014 and at 9500′ level in 2012 (Supplementary Table 2). These samples were stored in a glass vial sealed with a butyl rubber septum. Evidence from bulk carbon, hydrogen isotope signatures, and clumped methane has demonstrated that hydrocarbons of abiotic origin are predominant in Kidd Creek[10,41,62].

**Amadeus Basin, Australia**. Gas samples from Dingo gas field in Amadeus Basin were collected as described in previous study[65]. Both carbon and hydrogen signatures imply that hydrocarbons from Dingo gas field are derived from abiotic origins[12].

### Evaluation of the combinatorial effect of thermogenic ethane
We used the theoretical model to account for the carbon isotope composition of natural hydrocarbon gases produced by thermal cracking[21–23]. Various alkyl groups attached to a large kerogen molecule are assumed to produce hydrocarbon gases. In this model, the carbon atoms of any individual natural gas hydrocarbon molecule are defined as '$C_n$'. The carbon atom arising from the C-C bond breaking of

the alkyl chain is defined as '$C_m$'. The remaining C-atoms in the hydrocarbons are defined as '$C_p$'. Considering only primary isotope effects, the carbon atom of $C_m$ is enriched in $^{12}C$ because of the C-C bond breaking. However, the C-atoms ($C_p$) are unaffected by any isotope fractionation, thus recording the original isotope composition of the alkyl chain in kerogen. A given hydrocarbon with $n$ atoms has one $C_m$ atom and $(n-1)$ $C_p$ atoms. Therefore, its carbon isotope composition is:

$$\delta^{13}C_n = -\Delta_q/n + \delta^{13}C_p \tag{14}$$

If the $\delta^{13}C_n$ values are plotted as a function of $1/n$ ('natural gas plot' of ref. 21), the slope and the intercept of this plot represent the $\Delta_q$ (= $\delta^{13}C_p - \delta^{13}C_m$) and the $\delta^{13}C_p$ values, respectively, based on Eq. (14). The slope of the plot represents the isotope fractionation associated with C-C bond breaking, leading to the difference in the carbon isotope composition between two carbon atoms of ethane ('intramolecular bias'). Based on this model, the ethane produced contains two symmetrically equivalent carbon atoms, but they originate from precursor sites with different kinetic isotope effects during ethane production. The $^{13}C/^{12}C$ ratio of ethane ($R_{AVE}$) is the average of the two carbon positions, and because of the symmetry of ethane, the probability of $^{13}C_2H_6$ formation is expected to be proportional to the square of that average ratio ($R_{stochastic}$). In practice, however, the probability of the formation of $^{13}C_2H_6$ ($R_{26/24}$) is proportional to the product of the $^{13}C/^{12}C$ ratio of the two different carbon positions. The deviation of the $^{13}C$-$^{13}C$ isotopologues abundance ratio of ethane from stochastic ($\Delta^{13}C^{13}C_{Comb}$) is expressed as follows:

$$\Delta^{13}C^{13}C_{Comb} = 1000(R_{26/24}/R_{AVE}^2 - 1) \tag{15}$$

$$R_{AVE} = (R_A + R_B)/2 \tag{16}$$

$$R_{26/24} = R_A \times R_B \tag{17}$$

where $R_A$ and $R_B$ represent the $^{13}C/^{12}C$ ratio for the two carbon positions $C_A$ and $C_B$ of ethane, respectively. The solution to the simultaneous equations of Eqs. (16) and (17) can be expressed as:

$$R_A = R_{AVE} + (R_{AVE}^2 - R_{26/24})^{0.5} \tag{18}$$

$$R_B = R_{AVE} - (R_{AVE}^2 - R_{26/24})^{0.5} \tag{19}$$

Therefore, the difference in the $^{13}C/^{12}C$ ratio between the two carbon positions of ethane can be expressed as:

$$R_A - R_B = 2R_{AVE}(1 - R_{26/24}/R_{AVE}^2)^{0.5} \tag{20}$$

Converting Eq. (20) to δ values with respect to the standard ($R_{std}$) is expressed as:

$$\Delta^{13}C_{Ethane} = 2 \times (\delta^{13}C_{Ethane} + 1000) \times (-\Delta^{13}C^{13}C_{Comb}/1000)^{0.5} \tag{21}$$

where $\Delta^{13}C_{Ethane}$ and $\delta^{13}C_{Ethane}$ represent the difference and average of the two carbon positions in $\delta^{13}C$ between the two carbon positions of ethane. Equation (21) can be rearranged as:

$$\Delta^{13}C^{13}C_{Comb} = -(\Delta^{13}C_{Ethane}/2(\delta^{13}C_{Ethane} + 1000))^2 \times 1000 \tag{22}$$

Combinatorial isotope effects associated with intramolecular bias in the organic precursor can be estimated through the position-specific isotope composition of long-chain alkanes measured by nuclear magnetic resonance[66]. The latter shows differences in $\delta^{13}C$

values between two adjacent positions (=$\delta^{13}C_{CH3} - \delta^{13}C_{CH2}$) of ca. −3.9‰ (the $C_{16}$-$C_{31}$ range with odd carbon number), 10.4‰ (the $C_{16}$-$C_{31}$ range with even carbon number), and −12.5‰ (the $C_{11}$-$C_{15}$ range with odd and even carbon number)[66], which corresponds to depletion of $\Delta^{13}C^{13}C$ values of −0.004‰, −0.03‰, and −0.04‰, respectively. The combinatorial effects calculated here are much lower than the analytical precision of ±0.09‰ and can thus be quantitatively neglected.

## Data availability

All data generated or analysed during this study are included in this published article (and its supplementary information files).

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

## Acknowledgements

This work was supported by the Japan Society for the Promotion of Science KAKENHI grant JP21J22057 (K.T.), JP17H01165 (Y.U.), JP21H01198 (A.G.), and JP21K18646 (A.G.). Kidd Creek samples were collected via funding from the Natural Sciences and Engineering Research Council of Canada Discovery Grant (B.S.L). Further support from Canadian Institute for Advanced Research (CIFAR) is acknowledged for B.S.L as a Fellow of the Earth 4D Subsurface Science and Exploration Program. Justin Davies, Australia's Nuclear Science and Technology Organisation (ANSTO), Australia, is thanked for managing the gamma-ray irradiation of the methane experiment. Dianne Edwards and Emmanuelle Grosjean reviewed a pre-submission version of the paper, leading to many improvements. C.J.B publishes with the permission of the CEO, Geoscience Australia. Financial support was provided by the U.S. Department of Energy Geosciences program DE-SC0016271 (J.H.).

## Author contributions

K.T., Y.U., and A.G. conceived the project. K.T. and A.G. performed the gas isotopic analysis. K.T., Y.U., and A.G. managed the project and prepared the first draft of the manuscript. K.T. and A.G. collected the samples from Tokamachi mud volcano. B.S.L., T.G., C.J.B., and J.H. provided natural and experimental gas samples. Q.L. conducted theoretical calculations to estimate equilibrium distributions of $^{13}C$ isotopologues. All authors contributed to the final version of the manuscript.

## Competing interests

The authors declare no competing interests.
