## [Peer Review File · Nature Communications]

Low ^{13}C - ^{13}C abundances in abiotic ethaneREVIEWER COMMENTS

Reviewer #1 (Remarks to the Author):

This is a substantial, interesting and potentially impactful study that certainly deserves to be published in some form, but that has two substantial problems that must be fixed, as well as a variety of smaller issues that could benefit from another round of edits.

Perhaps the greatest weakness of the paper is the approach it has taken to prior work on the subject. If this just amounted to a missed reference I would note so in the line edits below and leave it at that, but the authors have tip-toed around something more substantial than that. This paper makes no reference to a prior paper on the 'clumped isotope' geochemistry of ethane, Clog et al., 2018 (results of which are also discussed in Eiler et al., 2018). In addition to the obvious question of scholarship, this means that the authors have not considered evidence outside their own analytical and experimental work, including a substantial amount of data for natural samples and experiments that suggest various processes relevant to production and destruction of thermogenic ethane may drive clumped isotope variations that extend over a range comparable to or larger than that seen in the author's work on abiogenic ethane. I.e., it is possible this prior work provides evidence for alternative processes that could be responsible for the signatures that are the main focus of Taguchi et al.'s interpretations, or evidence that abiogenic ethane overlaps compositions of some thermogenic ethanes, which would change the tenor of Taguchi et al.'s conclusions.

This prior study used a different analytical method than Taguchi et al. have used, and I imagine that they could have reasons to believe they have a more authoritative data set. And perhaps they will ultimately be proven to be right (or perhaps not). Be that as it may, simply ignoring prior and potentially contradictory evidence is not how we are supposed to be playing this game. I can't imagine how this paper could be published without considering and discussing such directly relevant prior work.

A second issue that stretches through most sections of this paper concerns the reference frame for the stochastic distribution of carbon isotopes in ethane. The authors clearly understand that the measurements they have made constrain relative differences in clumped isotope composition but do not anchor those measurements to an independent reference frame, such as the stochastic distribution that is commonly used in clumped isotope geochemistry. This is not an insurmountable problem, and in fact is also true of Clog et al.'s prior work on this subject, and of number of other clumped isotope studies that examine molecules that are challenging to drive to thermodynamic equilibrium. So, Taguchi et al.'s approach is perfectly within the community standards in this field. But it does limit the strength of some arguments one can base on such data, e.g., passages around lines 160-200 of this paper's discussion.

Taguchi et al. present an argument in favor of a preferred interpretation of the true clumped isotope composition of their working standard, and thus a way of 'anchoring' their data to the stochastic

distribution. However, I found the argument to be indirect, abstruse and speculative. It also might be quantitatively wrong: I believe the authors assume that the $\Delta^{13}\text{C}_{13}\text{C}$ value for ethanol that has reached intramolecular isotopic equilibrium will translate directly to that same value for ethane or CF_6 produced by chemical transformation of ethanol. This is untrue: equilibrated ethanol possesses a substantial site-specific difference in $\delta^{13}\text{C}$ between its methyl and CH_2OH moieties. I think this means that after ethanol is converted to ethane or CF_6 , where the two carbon positions are indistinguishable, the final $\Delta^{13}\text{C}_{13}\text{C}$ value should be something close to the sum of ethanol's equilibrium $\Delta^{13}\text{C}_{13}\text{C}$ value (correctly defined with knowledge of its site-specific structure) and the 'combinatorial' effect that arises when you can't distinguish two isotopically different sites. Thus, even if all of the other guesswork about the isotopic structure of biogenic ethanol were correct, I think the CF_6 produced from it would be measurably lower in $\Delta^{13}\text{C}_{13}\text{C}$ than the authors are guessing (unless I've misunderstood the details of their argument, which is possible – it is rather complex!).

The correct way to have approached this might have been to follow their suspicions that spark-discharge and gamma irradiation experiments drive ethane to equilibrium, by conducting a series of experiments that would hopefully show they could use one of these processes to create a time-independent, bracketed $\Delta^{13}\text{C}_{13}\text{C}$ value. If they had done this, it would have both proven their hypothesis regarding the isotopic effects of these processes (an important part of the discussion), while also providing a good estimate of the stochastic or high-temperature equilibrium clumped isotope reference frame. This study is publishable without such experiments, but if I were the authors I would very much want to do them first.

A related point: As far as I can tell, the authors have not measured clumped isotope compositions of experimental products that have highly predictable changes in $\Delta^{13}\text{C}_{13}\text{C}$, such as residues of diffusion or controlled mixtures of gases. This is how many prior studies have demonstrated at least relative accuracy of a clumped isotope method in the absence of, or in addition to, a demonstrably equilibrated reference frame (e.g., this has been done for CO_2 , N_2O , CH_4 , O_2 , H_2 , and the prior work on ethane). This sort of experiment doesn't let one assert an absolute or stochastic reference frame, but does lend confidence in one's measurements of relative differences in clumped isotope composition.

I've also made a variety of comments or questions concerning narrower points, keyed to the relevant line and figure numbers in the submitted manuscript:

105-106: Unclear; you seem to be alluding to KIE's associated with destruction of ethane, though the methods section only discusses the more common scenario in thermogenic gases – KIE's associated with production of ethane, which will have the opposite of the stated effect.

109: 'equal in position' has an ambiguous meaning; better to use a more formal statement of their symmetric equivalence.

110-113: This explanation misses the essential detail that in this passage we are discussing two (or more) positions that are symmetrically equivalent. In the case of symmetrically non-equivalent atomic sites, the definitions of site-specific isotope effects and the stochastic reference frame will take this into account and no 'combinatorial' effect will be observed (assuming no mistakes are made in the application of these concepts).

103-139: this section is somewhat repetitive. It also covers a variety of issues and models that were explained in greater detail in Clog et al., 2018 and Eiler et al., 2018. This section also neglects secondary cracking of ethane – something the data in Clog et al. 2018 suggest might be important to thermogenic gas suites.

163: An argument that presumes the samples can be placed in the stochastic reference frame.

175: An argument that presumes the samples can be placed in the stochastic reference frame.

199: Similar or greater relative $\Delta^{13}\text{C}_{13\text{C}}$ decreases were observed in several forms of thermogenic gases in Clog et al., 2018.

202-204: The meaning of 'extrinsic energy' is unclear; and the authors should explain why, specifically, they imagine ethane destruction leads to equilibrium. That is only true if the chemistry is reversible, at the level of elementary reactions, or is part of a cycle of individually irreversible reactions that create and destroy ethane at a steady state. The stipulation of a steady state is essential for these arguments; if the system evolves through cyclical but unbalanced reactions (i.e., with net production or consumption), then there is no clear argument to be made that you are moving toward equilibrium.

207: Are the authors proposing these are three body reactions? Doesn't this influence the reduced mass argument in the preceding section?

212-214: Not obvious what is meant here; is this some sort of total energy output, or per-photon energy content, or something else? How could one say spark discharge is higher energy than UV radiation if one doesn't specify these sorts of details?

213-228: As far as I can tell, this passage is conjectural; what evidence is there that any of this chemistry is taking place in the experiments? Perhaps that could be acceptable as a speculative hypothesis, but the

wording here suggests the authors know this chemistry is occurring through some independent observation or constraint.

Figure 2b: Why does the back-reaction process modify the slope of the CSIA trend, in the context of the model discussed? Are the vectors for these processes notional, or do they reflect a real model calculation of the coupled change in the CSIA slope and ethane clumped isotope index?

388-390: The slope is an empirical observation; the association of it with a particular pattern of site-specific isotope effect is a model interpretation. The word 'corresponds' doesn't really capture this ambiguity.

445: 'Scrambling' is not clearly described; does it refer here to ion source fragmentation and recombination, as is well documented for CO₂, N₂O, O₂, etc., or to some form of chemical exchange that accompanies the fluorination process?

445-448: The language here is vague and difficult to follow; I feel I understand these issues well but come away confused as to when the authors are discussing a stochastic reference frame and when they are discussing a difference in $\Delta^{13}\text{C}_{13}\text{C}$ from an arbitrary working standard. It seems clear the authors basically understand the issues, and I'm sure I do, but somehow the text is confusing anyway.

487: Why would one use CH₂OH-CH₂OH as a stand-in for ethanol (CH₃-CH₂OH)?

657: Awkwardly put; I follow what is being done here, but the last half of this sentence does not convey a clear meaning.

678-680: Not clear; when this difference is zero (clearly within the given range between negative and positive the contrast will be zero, not within the range -0.2 ± 0.1).

Reviewer #2 (Remarks to the Author):

Taguchi et al present a very interesting study on carbon isotope clumping in ethane and its implications for understanding ethane sources and sinks. The paper provides new insights into what processes can be inferred from clumped isotope variation, and how they can be distinguished from one another. I think the paper will have important implications for ethane (bio)geochemistry, and for advanced organic isotope geochemistry more generally. This is a frontier area of isotope and organic geochemistry, and I think the results are exciting and worthy of publication in Nature Communications. While the results do not have specific implications for any one application of this measurement, they point the way forward for future work in a number of areas including natural gas geochemistry, deep-sea biogeochemistry (i.e. ethane oxidation), and astrobiology/origin of life studies (i.e. abiotic hydrocarbon formation). Given the value to all of these fields, I think the paper merits publication in this journal.

I am not familiar with the fluorination method used in this study, having more experience with high resolution isotope mass spectrometry of intact molecules (mostly methane, tangential experience with ethane). Therefore I can't comment directly on the methods, though having read the previous methods paper they seem robust. The inability to calibrate data to a stochastic reference frame is somewhat problematic for the long-term development of this technique, but not really a critique of this paper.

I think the paper should be published with minor revisions. I have a few line by line comments below. Generally speaking, I think some aspects of the paper need more elaboration and clarity, space permitting.

Sincerely,

Peter Douglas

Line by line comments

L46: More robust information is vague- is it possible to be more specific?

L48: Is the abiotic signature a certainty? Maybe change to 'likely abiotic'

L52: From my reading below, it is difficult to place these values on a working gas reference frame in a stochastic reference frame. How confident can you be these values are 'near stochastic'? Maybe add 'presumably' to indicate this uncertainty?

L71: move 'ratios' to after 2H/1H

L92: should be 'exhibits'

L130: This is a large uncertainty. Please clarify (maybe in the caption for Figure 2a) that this uncertainty is included in the error envelope.

L135: this estimate based on bio-ethanol is very important, and it is important to verify the value measured is representative. It might be good to clarify how many samples have been analyzed, what they represent in terms of different plant sources, and how much variability they demonstrated.

L186: I don't understand why this ratio of kinetic isotope effects, if they follow a stochastic distribution, should be equal to zero. More explanation here or in the supplement, and/or a citation, is needed.

L225: Clarify this or add more citations. What would cause 'variable' reversibility in these reactions? Under what circumstances would you expect them to be more or less reversible? If the reversibility is variable, its not clear that repeated cleavage and formation of the bonds would trend towards an equilibrium composition. Please expand on this prediction to further justify it, either here or in the supplement.

L231: The 'increase in D13C13C' referred to here is not clear. Increase as a function of what other variable? I'm assuming it's the slope of the CSIA trend, but this is not clear here.

L238: How long would the proposed isotope exchange take? Has it been modelled? If so, is the timescale reasonable for these gases? Recent research on methane suggests that hydrogen isotope exchange can take a very long time (at least at low temperatures in the absence of catalysts) (Turner et al, 2022, GCA). What is the the estimated age of production of these gases?

L247: Mars is always an exciting target, but I'm not sure this method is feasible for application to Mars, so it seems kind of odd to discuss it here. Presumably this measurement isn't going to happen with by Rover, so would it entail a gas sample return mission? Has anyone proposed this? I'd say if it isn't realistic in the next few decades it probably shouldn't be discussed as an implication of the research.

Fig 2. What does it mean that the natural gas data are aligned perpendicular to the temperature curve? Is there another process that could induce variation other than the temperature of formation? I think this deserves some discussion.

L444: How is this lambda value known? Maybe a citation is necessary here, or more elaboration.

L447: I think I understand this sentence, but it is written awkwardly. I think it's saying that we can only be confident in relative values between samples, not absolute values. Please re-word.

L612: I think it would be good to provide more details on previous evidence of ethane oxidation in these samples. Is there independent evidence indicating this process at this site? Based on other studies of ethane oxidation, is this a likely location? It would be nice to see independent corroboration.

Reference cited

Turner, Andrew C., et al. "Experimental determination of hydrogen isotope exchange rates between methane and water under hydrothermal conditions." *Geochimica et Cosmochimica Acta* (2022).

Reviewer #3 (Remarks to the Author):

This manuscript reports ethane clumped isotope data that show promising patterns for distinguishing abiotic and biotic ethane. The type of data presented here are of high interests to people in the field, and the implications should be quite impactful to the general readership of *Nature Communications*. I have a few suggestions that are mostly minor, except for some concerns about the standardization process. Pending these revisions, I would recommend publication.

L52: It is too parsimonious to attribute thermogenic ethane's near-stochastic clumping to biological precursors. In thermogenic ethane production, clumped isotope signatures could be a result of many factors: 1) KIE and statistical effects in ethane generation 2) precursor isotope distribution 3) KIE and statistical effects in ethane destruction (Peterson et al. 2018). 4) potential equilibration at higher

maturites. The near stochastic distribution could be a result small effects in all of these categories, or nuanced cancellation these effects.

L80-81: More durable is too general. In thermal cracking, C—C bonds are much easier to break than C—H bonds. Please state what types of chemistry C—C bonds are more resistant to specifically.

L86: “and compared them with”

L97: The methods of lab synthesis of abiotic gas do not include FTT. This is a little regretful, because FTT is more relevant for abiogenesis on Earth environments. I suggest that the authors give a brief overview of different abiotic synthesis in the introduction and describe the scope of this study, because the introduction only talks about FTT. This is also a big distinction that the broad science readership of nature communications might be unfamiliar with.

L98: “can be” should be “are”

L127-128: These parameters can be moved into the Methods.

L164–165: The highest $\Delta^{13}\text{C}_{13\text{C}}$ sample from Tokamachi corresponds to an equilibrium temperature well below 0C, making equilibration unlikely.

L188–191: That’s right. It is hard to explain why the UV experiments have $\Delta^{13}\text{C}_{13\text{C}}$ around -0.9 per mil, while the prediction of this collision frequency theory is at -0.52 per mil. Although it could possibly arise from the error in absolute reference frame.

L207: If you add calculation of the collision KIE with the third body M, would the clumped isotope effect shift much?

L244–247: Titan is a more relevant place for the photolysis experiments studied in this work. Titan’s organic synthesis also starts from methane like in this study.

L462–465: “the aldolization and ketolisation reactions are partially at equilibrium(44).” and “These C-C bonds are produced in an aldolisation reaction, which is thought to be at equilibrium.” These two sentences are conflicting as the first one says partial equilibrium but second says full equilibrium. Besides, It is unclear to me how C-5 and C-6 positions are controlled by the reversible reaction of aldolase, because Gilbert et al. (2013) only applied equilibrium analysis to C-3 and C-4 positions. Please elaborate. Another caveat of this treatment is that there might be KIE in ethanol fermentation, unless the authors could prove that the reaction is non-fractionating or quantitative.

L469: The difficulty to perform DFT calculations on glucose is not so much higher than doing so on C2 molecules...

L810–811: Just say “eq. 9–10”

Comment and Reply

REVIEWER COMMENTS

Reviewer #1 (Remarks to the Author):

This is a substantial, interesting and potentially impactful study that certainly deserves to be published in some form, but that has two substantial problems that must be fixed, as well as a variety of smaller issues that could benefit from another round of edits.

Perhaps the greatest weakness of the paper is the approach it has taken to prior work on the subject. If this just amounted to a missed reference I would note so in the line edits below and leave it at that, but the authors have tip-toed around something more substantial than that. This paper makes no reference to a prior paper on the ‘clumped isotope’ geochemistry of ethane, Clog et al., 2018 (results of which are also discussed in Eiler et al., 2018). In addition to the obvious question of scholarship, this means that the authors have not considered evidence outside their own analytical and experimental work, including a substantial amount of data for natural samples and experiments that suggest various processes relevant to production and destruction of thermogenic ethane may drive clumped isotope variations that extend over a range comparable to or larger than that seen in the author’s work on abiotic ethane. I.e., it is possible this prior work provides evidence for alternative processes that could be responsible for the signatures that are the main focus of Taguchi et al.’s interpretations, or evidence that abiotic ethane overlaps compositions of some thermogenic ethanes, which would change the tenor of Taguchi et al.’s conclusions. This prior study used a different analytical method than Taguchi et al. have used, and I imagine that they could have reasons to believe they have a more authoritative data set. And perhaps they will ultimately be proven to be right (or perhaps not). Be that as it may, simply ignoring prior and potentially contradictory evidence is not how we are supposed to be playing this game. I can’t imagine how this paper could be published without considering and discussing such directly relevant prior work.

We thank the reviewer for reminding us of an important point. We regret this oversight and agree with the reviewer. The paper by Clog et al. was a strong inspiration for our work. Personally, we have been discussing the issue with Dr. Clog himself in an international conference. What happened is that we focused on the interpretation of the data within our own frame. Unfortunately, after revising the manuscript into the length within the page limit, the Clog et al paper was dropped from the main text.

We really regret this situation and have changed the manuscript as explained below. In particular, we discuss extensively, both in the manuscript and in the supplementary material, the discrepancies between our data and that obtained by Clog et al.

The main discrepancy relies in the range of variation observed by Clog et al. (over 4‰) which is ca. 6 times higher than the variations observed in our study (0.57‰). The explanation for this discrepancy has no explanation yet. In addition, we could not, unfortunately, measure the same samples measured in Clog et al. which makes it even more difficult to compare the data. The only comparison we can make is with the data coming from the pyrolysis experiment. Clog et al. found a depletion of –1.69‰ in the remaining ethane compared to the starting ethane while we found no significant change in the $\Delta^{13}\text{C}^{13}\text{C}$ value throughout the course of ethane pyrolysis in the same conditions. This is consistent with the lower range of values apparently given by our method. However, the scale compression was precisely checked in Taguchi et al 2021. At the same time, we do not see any explanation for an enlargement of the scale from

Clog's method. To summarize, this methodological problem is yet to be solved. While this is undoubtedly a matter that must be investigated in future studies, especially given the potential of ^{13}C - ^{13}C clumping for organic geochemistry, we still think this is out of the scope of the present paper. Most importantly, this does not change the conclusions made in the present study since the data have been obtained by the same method.

To clarify, we have added the data from the pyrolysis experiment to Supplemental Information and have added the following text to the General Trends (L100-L109). We have also added the reference Clog et al., 2018 and Eiler et al., 2018 to the list of references. We believe that these modifications adequately address the reviewer's comment.

Added text,

“A previous study using high-resolution mass spectrometry exhibited 4‰ variation of $\Delta^{13}\text{C}^{13}\text{C}$ values of thermogenic ethane²⁴. This variation has been suggested to arise from the pyrolysis of ethane, which leads to a decrease in $\Delta^{13}\text{C}^{13}\text{C}$ values^{24,25}. The data obtained by our method show a narrower range of the $\Delta^{13}\text{C}^{13}\text{C}$ values (0.57‰). A pyrolysis experiment conducted in the same conditions showed no change in the $\Delta^{13}\text{C}^{13}\text{C}$ values (see SI). These observations point to a discrepancy between the two methods, the reason for which is currently unknown and will necessitate further interlaboratory comparisons. Presently, the comparison of data between the two methods would thus be inaccurate. Hence, while we consider the calibration of both methods an important problem to solve in the future, the data presented here will be that obtained by the method presented in ref²².”

Added reference,

25. Clog M. et al. A reconnaissance study of ^{13}C - ^{13}C clumping in ethane from natural gas. *Geochim. Cosmochim. Acta* **223**, 229–244 (2018). <https://doi.org/10.1016/j.gca.2017.12.004>.
26. Eiler, J. M. et al. The isotopic structures of geological organic compounds. *Geol. Soc.* **468**, 53–81 (2018). <https://doi.org/10.1144/SP468.4>.

A second issue that stretches through most sections of this paper concerns the reference frame for the stochastic distribution of carbon isotopes in ethane. The authors clearly understand that the measurements they have made constrain relative differences in clumped isotope composition but do not anchor those measurements to an independent reference frame, such as the stochastic distribution that is commonly used in clumped isotope geochemistry. This is not an insurmountable problem, and in fact is also true of Clog et al.'s prior work on this subject, and of number of other clumped isotope studies that examine molecules that are challenging to drive to thermodynamic equilibrium. So, Taguchi et al.'s approach is perfectly within the community standards in this field. But it does limit the strength of some arguments one can base on such data, e.g., passages around lines 160-200 of this paper's discussion. Taguchi et al. present an argument in favor of a preferred interpretation of the true clumped isotope composition of their working standard, and thus a way of 'anchoring' their data to the stochastic distribution. However, I found the argument to be indirect, abstruse and speculative. It also might be quantitatively wrong: I believe the authors assume that the $\Delta^{13}\text{C}^{13}\text{C}$ value for ethanol that has reached intramolecular isotopic equilibrium will translate directly to that same value for ethane or CF_6 produced by chemical transformation of ethanol. This is untrue: equilibrated ethanol possesses a substantial site-specific difference in $\delta^{13}\text{C}$ between its methyl and CH_2OH moieties. I think this means that after ethanol is converted to ethane or CF_6 , where the two carbon positions are indistinguishable, the final $\Delta^{13}\text{C}^{13}\text{C}$ value should be something close to the sum of ethanol's equilibrium $\Delta^{13}\text{C}^{13}\text{C}$ value (correctly defined with knowledge of the its site-specific structure) and the 'combinatorial' effect that arises when you can't distinguish two isotopically different sites. Thus, even if all of the other guesswork about the isotopic

structure of biogenic ethanol were correct, I think the CF₆ produced from it would be measurably lower in $\Delta^{13}\text{C}^{13}\text{C}$ than the authors are guessing (unless I've misunderstood the details of their argument, which is possible – it is rather complex!).

Good point. As the reviewer suggested, $\Delta^{13}\text{C}^{13}\text{C}$ may potentially change during the conversion from ethanol to C₂F₆, though the expected change should be small enough to neglect in a current analytical precision of the $\Delta^{13}\text{C}^{13}\text{C}$ value. Thus, this issue does not change the main arguments of this paper.

The combinatorial effect arises from a statistical clumped isotope anomaly that occur in any system when two atoms of the same element in a molecule are indistinguishable, but with different isotopic composition. We agree with the reviewer: although the two carbon atoms in ethanol are distinguishable and have different isotope composition (the difference between two carbon atom is up to 11.5‰; Taguchi et al., 2020; Gilbert et al. 2013), we measured $\Delta^{13}\text{C}^{13}\text{C}$ of ethanol as C₂F₆ where the two carbon atoms become equivalent by symmetry. Therefore, the $\Delta^{13}\text{C}^{13}\text{C}$ value of ethanol obtained here must be affected by a combinatorial isotope effect. However, a quantitative estimation of the latter gives an expected combinatorial isotope effect of ca. -0.03‰ (calculated using the formula of Rockman et al. 2016) at maximum which is lower than analytical precision of $\pm 0.09\%$. Thus, a combinatorial effect in ethanol must arise from the isotope heterogeneity of ethanol, but the effect is quantitatively negligible.

To clarify, we have added the reference and the following text to the Method. We believe that this new addition adequately addresses the reviewer's comment.

Added text,

L533-L539 :“In natural ethanol, the $\delta^{13}\text{C}$ values in positions CH₃ and CH₂OH can be different by up to 11.4‰, which would result in lower $\Delta^{13}\text{C}^{13}\text{C}$ values compared with the stochastic distribution owing to combinatorial effects²⁸ (see Main text). However, the combinatorial effect in ethanol measured in this study is up to -0.03‰ associated with site-specific ¹³C distribution in ethanol of 11.4‰ at maximum⁵⁷. The combinatorial effects calculated here is much lower than the analytical precision of $\pm 0.09\%$ and can thus be quantitatively neglected here.”

Added reference,

57. Gilbert, A., Yamada, K. & Yoshida, N. Accurate method for the determination of intramolecular ¹³C isotope composition of ethanol from aqueous solutions. *Anal. Chem.* **85**, 6566–6570 (2013). <http://doi.org/10.1021/ac401021p>.

The correct way to have approached this might have been to follow their suspicions that spark-discharge and gamma irradiation experiments drive ethane to equilibrium, by conducting a series of experiments that would hopefully show they could use one of these processes to create a time-independent, bracketed $\Delta^{13}\text{C}^{13}\text{C}$ value. If they had done this, it would have both proven their hypothesis regarding the isotopic effects of these processes (an important part of the discussion), while also providing a good estimate of the stochastic or high-temperature equilibrium clumped isotope reference frame. This study is publishable without such experiments, but if I were the authors I would very much want to do them first.

We thank the reviewer for this insightful comment. We actually did two spark-discharge experiments with different times (15 minutes and 5 hours). The $\Delta^{13}\text{C}^{13}\text{C}$ values did not show any change between the two experiments. We suggest that the destruction of hydrocarbon after its production enhance back reaction in CH₃ radical recombination and drives the ¹³C-¹³C abundance in ethane to partial isotopic equilibrium. To reach full isotopic equilibrium, the C-C bond formation and destruction reaction rate should be equal to each other. However, the two spark discharge experiments show accumulation of

ethane concentration but no $\Delta^{13}\text{C}^{13}\text{C}$ variations, indicating that the isotopic composition in ethane is not fully equilibrated. Therefore, the $\Delta^{13}\text{C}^{13}\text{C}$ may not reach the homogeneous isotopic equilibrium completely, but only partially. We are planning to run experiments starting from ethane with different starting $\Delta^{13}\text{C}^{13}\text{C}$ values to take advantage of the partial re-equilibration to constrain further the stochastic value, though more detailed mechanisms are beyond the scope of this paper.

A related point: As far as I can tell, the authors have not measured clumped isotope compositions of experimental products that have highly predictable changes in $\Delta^{13}\text{C}^{13}\text{C}$, such as residues of diffusion or controlled mixtures of gases. This is how many prior studies have demonstrated at least relative accuracy of a clumped isotope method in the absence of, or in addition to, a demonstrably equilibrated reference frame (e.g., this has been done for CO_2 , N_2O , CH_4 , O_2 , H_2 , and the prior work on ethane). This sort of experiment doesn't let one assert an absolute or stochastic reference frame, but does lend confidence in one's measurements of relative differences in clumped isotope composition.

We appreciate the reviewer's comment and suggestion. In this study, we rely on our previous work using controlled spiked experiment of ^{13}C - ^{13}C labelled ethanol, which showed that relative measured by our method seems accurate (Taguchi et al., 2021). However, given the discrepancy between the Clog's paper and our work, the diffusion experiment will be insightful. In this manuscript, we have cited our previous work (Taguchi et al., 2021) for the study of controlled mixture of gases.

I've also made a variety of comments or questions concerning narrower points, keyed to the relevant line and figure numbers in the submitted manuscript:

105-106: Unclear; you seem to be alluding to KIE's associated with destruction of ethane, though the methods section only discusses the more common scenario in thermogenic gases – KIE's associated with production of ethane, which will have the opposite of the stated effect.

Done. For clarity, we have changed the manuscripts as follow (at L114-L116):

Original text,

“The kinetic isotope effect is relevant only to one carbon in ethane, which undergoes cracking, resulting in preferential ^{13}C enrichment in one of the two carbons.”

↓

Revised text,

“In an ideal case where ethane is produced by breaking at least one C-C bond in an organic precursor, the kinetic isotope effect is relevant only to one carbon in ethane resulting in preferential ^{13}C enrichment in one of the two carbons.”

109: ‘equal in position’ has an ambiguous meaning; better to use a more formal statement of their symmetric equivalence.

Done. We have changed “equal in position” to “symmetrically equivalent” (at L118).

110-113: This explanation misses the essential detail that in this passage we are discussing two (or more) positions that are symmetrically equivalent. In the case of symmetrically non-equivalent atomic sites, the definitions of site-specific isotope effects and the stochastic reference frame will take this into account and no ‘combinatorial’ effect will be observed (assuming no mistakes are made in the application of these concepts).

Done. We have added a sentence to clarify as follow:

Added text,

L122-L124: “This does not apply to molecules with non-equivalent atomic sites, typically ethanol, for which an accurate stochastic distribution can be calculated based on the $^{13}\text{C}/^{12}\text{C}$ ratio of both sites.”

103-139: this section is somewhat repetitive. It also covers a variety of issues and models that were explained in greater detail in Clog et al., 2018 and Eiler et al., 2018. This section also neglects secondary cracking of ethane – something the data in Clog et al. 2018 suggest might be important to thermogenic gas suites.

Agreed. The sentence which might be repetitive was removed and we mentioned that a model was previously discussed in detail in Clog et al. (2018) and Eiler et al. (2018). We also discuss the fact that secondary cracking of ethane itself didn't contribute to thermogenic $\Delta^{13}\text{C}^{13}\text{C}$ values based on ethane pyrolysis experiment in this study.

Removed sentence,

L116-L118 in the original manuscript: “In the present model, only one carbon atom of the formed hydrocarbon is affected by the cracking-related isotope effect, with other atoms remaining unaffected.”

Added sentence,

L113-L114: “, as discussed in previous study (Clog et al., 2018, Eiler et al. 2018)”

Added sentence,

L189-L197: “Other processes potentially alter the $\Delta^{13}\text{C}^{13}\text{C}$ value in ethane and are compatible with the observed variation of thermogenic ethane. These include diffusion (increase of $\Delta^{13}\text{C}^{13}\text{C}$ value by 0.3‰ in the case where molecular collision is important), mixing with different sources (increase of $\Delta^{13}\text{C}^{13}\text{C}$ value by up to 0.13‰ in the case of mixing samples with the same $\Delta^{13}\text{C}^{13}\text{C}$ values but with different $\delta^{13}\text{C}$ values of –20‰ and –45‰) and secondary cracking of ethane itself (no $\Delta^{13}\text{C}^{13}\text{C}$ variations at 600°C in this study; see SI)^{25,26}. However, again, the discrimination potential of the $\delta^{13}\text{C}^{13}\text{C}$ value of ethane is not weakened, because all these factors tend to increase the $\Delta^{13}\text{C}^{13}\text{C}$ value.”

163: An argument that presumes the samples can be placed in the stochastic reference frame.

Yes. We added the sentence to clarify that stochastic reference frame is still hypothetical.

Added sentence,

L181-L182: “based on the assumption of stochastic reference frame (see Method).”

175: An argument that presumes the samples can be placed in the stochastic reference frame.

Yes. We have added the sentence accordingly (see comment just above).

Added sentence,

L200: “In a stochastic reference frame assumed here (see Method),”

199: Similar or greater relative $\Delta^{13}\text{C}^{13}\text{C}$ decreases were observed in several forms of thermogenic gases in Clog et al., 2018.

This comment was discussed in detail in the responses to the previous comments. We have added a paragraph in the section "General Trend" (L100-L109) in the revised manuscript.

202-204: The meaning of 'extrinsic energy' is unclear; and the authors should explain why, specifically, they imagine ethane destruction leads to equilibrium. That is only true if the chemistry is reversible, at the level of elementary reactions, or is part of a cycle of individually irreversible reactions that create and destroy ethane at a steady state. The stipulation of a steady state is essential for these arguments; if the system evolves through cyclical but unbalanced reactions (i.e., with net production or consumption), then there is no clear argument to be made that you are moving toward equilibrium.

Agreed. We have changed the manuscripts to clarify our hypothesis as follow (L227-L229):

Original text,

"After hydrocarbons are produced, exposure to extrinsic energy leads to their decomposition, with $\Delta^{13}\text{C}^{13}\text{C}$ of ethane approaching isotopic equilibrium (+0.22‰ at 25°C; Extended Data Fig. 2)."

↓

Revised text,

"After hydrocarbons are produced, its destruction enhances carbon exchange among individual hydrocarbons, which may potentially lead partial isotopic equilibrium through repeated production and destruction cycling."

207: Are the authors proposing these are three body reactions? Doesn't this influence the reduced mass argument in the preceding section?

Yes. The recombination reaction between CH_3 radical is via the formation of C_2H_6^* with high energy (R1) and deactivation to C_2H_6 by its collision with another molecule/atom (third body) M (R2).

The pressure of our experiments (about 12kPa of CH_4) is range of high-pressure-limit, leading to an almost quantitative conversion of C_2H_6^* to C_2H_6 . Therefore, the third body, M, does not contribute to the isotope effect during the experiment here. For explaining details, we have added the following section to the Supplementally Information:

Added section in Supplementally Information,

"The recombination reaction between CH_3 radical is via the formation of C_2H_6^* with high energy (S10) and deactivation to C_2H_6 by its collision with another molecule/atom (third body) M (S11) as follow:

The pressure of our experiments (about 12kPa of CH_4) is range of high-pressure-limit, leading to an almost quantitative conversion of C_2H_6^* to C_2H_6 . Therefore, the third body, M, does not contribute to the isotope effect during the experiment here."

212-214: Not obvious what is meant here; is this some sort of total energy output, or per-photon energy content, or something else? How could one say spark discharge is higher energy than UV radiation if one doesn't specify these sorts of details?

The paragraph was rewritten (L237-L241 in the revised manuscript) as follow:

Original text,

“Hence, the C₂H₆ and C₃H₈ produced by UV irradiation of methane are unlikely to undergo C-C bond decomposition. Conversely, spark discharge and gamma-rays have more energy than UV irradiation; thus, the C-C bonds of C₂+ hydrocarbons are frequently cleaved after their formation as follows (Extended Data Table 4):”

↓

Revised text,

“Hence, the C₂H₆ and C₃H₈ produced by UV irradiation of methane are unlikely to undergo C-C bond decomposition because of the lack of high energy photon below 150 nm in our experimental setting. Conversely, for spark discharge and gamma-rays experiment, the C-C bonds of C₂+ hydrocarbons frequently cleave after their formation as follows⁴² (Extended Data Table 4):”

213-228: As far as I can tell, this passage is conjectural; what evidence is there that any of this chemistry is taking place in the experiments? Perhaps that could be acceptable as a speculative hypothesis, but the wording here suggests the authors know this chemistry is occurring through some independent observation or constraint.

This paragraph was rewritten (see comment above). Also, we have added the reference for the chemistry occurring in the similar experimental conditions at L241.

Added reference,

42. Kado, S. et al. Reaction mechanism of methane activation using non-equilibrium pulsed discharge at room temperature. *Fuel*. **82**, 2291-2297 (2003). [http://doi.org/10.1016/S0016-2361\(03\)00163-7](http://doi.org/10.1016/S0016-2361(03)00163-7).

Figure 2b: Why does the back-reaction process modify the slope of the CSIA trend, in the context of the model discussed? Are the vectors for these processes notional, or do they reflect a real model calculation of the coupled change in the CSIA slope and ethane clumped isotope index?

In the spark discharge and gamma irradiation experiments, we predict that the decomposition of hydrocarbons enhances back-reaction process with exchange of carbon between individual hydrocarbons. This reaction network changes not only the $\Delta^{13}\text{C}^{13}\text{C}$ of ethane, but also the $^{13}\text{C}/^{12}\text{C}$ ratio (CSIA trend) between the hydrocarbons. However, how the $\Delta^{13}\text{C}^{13}\text{C}$ of ethane and CSIA trend are controlled is difficult to model because of the complexity of radical reaction network and its isotope effect. Thus, the direction of arrow in Fig. 2b is notional. To clarify, we change the sentence “predicted” to “notional” at L449.

388-390: The slope is an empirical observation; the association of it with a particular pattern of site-specific isotope effect is a model interpretation. The word ‘corresponds’ doesn’t really capture this ambiguity.

Done. We have changed ambiguous wording as follow:

L435: “corresponds to” to “assumed to be”

445: ‘Scrambling’ is not clearly described; does it refer here to ion source fragmentation and recombination, as is well documented for CO₂, N₂O, O₂, etc., or to some form of chemical exchange that accompanies the fluorination process?

Done. We have modified the sentence for clarity as follow:

L491-L493: “which may have occurred during the fluorination of ethene but not in the ion source of the mass spectrometer.”

445-448: The language here is vague and difficult to follow; I feel I understand these issues well but come away confused as to when the authors are discussing a stochastic reference frame and when they are discussing a difference in $\Delta^{13}\text{C}^{13}\text{C}$ from an arbitrary working standard. It seems clear the authors basically understand the issues, and I’m sure I do, but somehow the text is confusing anyway.

For avoiding confusion and redundancy, we have deleted the two sentences in the revised manuscript (L445-L4447 in original manuscript).

Deleted text:

“Because of the absence of a stochastic frame, the Δ values discussed above are not relative to a stochastic distribution but to a working standard. Therefore, the comparison between the two samples is important in this study.”

487: Why would one use $\text{CH}_2\text{OH}-\text{CH}_2\text{OH}$ as a stand-in for ethanol ($\text{CH}_3-\text{CH}_2\text{OH}$)?

This is because the bio-ethanol is derived from $\text{CH}_2\text{OH}-\text{CH}_2\text{OH}$ position of the precursor glucose. We have changed a sentence to clarity as follow (L526-L529):

Original text,

“Thus, we used data obtained from $\text{CH}_2\text{OH}-\text{CH}_2\text{OH}$ to predict $\Delta^{13}\text{C}^{13}\text{C}$ of the C-C bonds in glucose under thermodynamic equilibrium.”

↓

Revised text,

“Thus, we used data obtained from $\text{CH}_2\text{OH}-\text{CH}_2\text{OH}$ to predict $\Delta^{13}\text{C}^{13}\text{C}$ of the C_1-C_2 and C_5-C_6 bonds in glucose under thermodynamic equilibrium, because the carbons in the C_1-C_2 and C_5-C_6 positions of glucose are composed as $\text{CH}_2\text{OH}-\text{CHOH}-$ and $\text{CHO}-\text{CHOH}-$, respectively.”

657: Awkwardly put; I follow what is being done here, but the last half of this sentence does not convey a clear meaning.

Done. Also, an error has been corrected in accordance with the reviewer's comment (L708-L709).

Original text,

“In practice, however, the probability of the formation of $^{13}\text{C}_2\text{H}_6$ is proportional to the $^{13}\text{C}/^{12}\text{C}$ ratio of the two carbon positions multiplied by their respective ratios ($R_{26/24}$).”

↓

Revised text,

“In practice, however, the probability of the formation of $^{13}\text{C}_2\text{H}_6$ ($R_{26/24}$) is proportional to the product of the $^{13}\text{C}/^{12}\text{C}$ ratio of the two different carbon positions.”

678-680: Not clear; when this difference is zero (clearly within the given range between negative and positive the contrast will be zero, not within the range -0.2 ± 0.1).

We have modified the sentence (L730-L736):

Original text,

“The latter show differences in $\delta^{13}\text{C}$ values between two adjacent positions ranging from -13.5‰ to 17.6‰ , which corresponds to depletion of $\Delta^{13}\text{C}^{13}\text{C}$ values of $-0.20 \pm 0.10\text{‰}$.”

↓

Revised text,

“The latter show differences in $\delta^{13}\text{C}$ values between two adjacent positions ($=\delta^{13}\text{C}_{\text{CH}_3} - \delta^{13}\text{C}_{\text{CH}_2}$) of ca. -3.9‰ (the $\text{C}_{16}\text{-C}_{31}$ range with odd carbon number), 10.4‰ (the $\text{C}_{16}\text{-C}_{31}$ range with even carbon number), and -12.5‰ (the $\text{C}_{11}\text{-C}_{15}$ range with odd and even carbon number), which corresponds to depletions of $\Delta^{13}\text{C}^{13}\text{C}$ values of -0.004‰ , -0.03‰ , and -0.04‰ , respectively. The combinatorial effects calculated here is much lower than the analytical precision of $\pm 0.09\text{‰}$ and can thus be quantitatively neglected.”

Reviewer #2 (Remarks to the Author):

Taguchi et al present a very interesting study on carbon isotope clumping in ethane and its implications for understanding ethane sources and sinks. The paper provides new insights into what processes can be inferred from clumped isotope variation, and how they can be distinguished from one another. I think the paper will have important implications for ethane (bio)geochemistry, and for advanced organic isotope geochemistry more generally. This is a frontier area of isotope and organic geochemistry, and I think the results are exciting and worthy of publication in Nature Communications. While the results do not have specific implications for any one application of this measurement, they point the way forward for future work in a number of areas including natural gas geochemistry, deep-sea biogeochemistry (i.e. ethane oxidation), and astrobiology/origin of life studies (i.e. abiotic hydrocarbon formation). Given the value to all of these fields, I think the paper merits publication in this journal.

I am not familiar with the fluorination method used in this study, having more experience with high resolution isotope mass spectrometry of intact molecules (mostly methane, tangential experience with ethane). Therefore I can't comment directly on the methods, though having read the previous methods paper they seem robust. The inability to calibrate data to a stochastic reference frame is somewhat problematic for the long-term development of this technique, but not really a critique of this paper.

I think the paper should be published with minor revisions. I have a few line by line comments below. Generally speaking, I think some aspects of the paper need more elaboration and clarity, space permitting.

Sincerely,

Peter Douglas

We are grateful to the reviewer's enthusiastic comments.

Line by line comments

L46: More robust information is vague- is it possible to be more specific?

Done. We have added the sentence as follows to clarify the manuscript (L45):

Original text,

“Recent development of clumped-isotope analysis provides more robust information, independent of the stable isotopic composition of the starting material.”

↓

Revised text,

“Recent development of clumped-isotope analysis provides more robust information, rely on the independence from the stable isotopic composition of the starting material.”

L48: Is the abiotic signature a certainty? Maybe change to ‘likely abiotic’

We have modified the sentence for clarity (L47)

Original text,

“that the abiotically produced ethane”

↓

Revised text,

“that the abiotically-synthesized ethane”

L52: From my reading below, it is difficult to place these values on a working gas reference frame in a stochastic reference frame. How confident can you be these values are ‘near stochastic’? Maybe add ‘presumably’ to indicate this uncertainty?

Done. We have added “presumably” at L51 to clarify that the stochastic reference frame we use is an assumption based on indirect evidence.

L71: move ‘ratios’ to after 2H/1H

Done.

L92: should be ‘exhibits’

Done.

L130: This is a large uncertainty. Please clarify (maybe in the caption for Figure 2a) that this uncertainty is included in the error envelope.

Disagreed. The difference between “-0.04‰ and 0.45‰” is not the error. The two values are the kinetic isotope effect at 25°C and at 300°C.

L135: this estimate based on bio-ethanol is very important, and it is important to verify the value measured is representative. It might be good to clarify how many samples have been analyzed, what they represent in terms of different plant sources, and how much variability they demonstrated.

Agreed. We have added the supporting sentence as follow (L144-L146):

“Despite of different type of photosynthetic pathway (C3, C4, and CAM plants), the bio-ethanol show a narrow range of $\Delta^{13}\text{C}^{13}\text{C}$ values, suggesting bio-ethanol as a good representative of biological molecules²².”

L186: I don’t understand why this ratio of kinetic isotope effects, if they follow a stochastic distribution, should be equal to zero. More explanation here or in the supplement, and/or a citation, is needed.

For explaining the details, we have added the following text to the main text and section to the Supplemental Information:

Added sentence to main text,
L211-L212: (see SI)

Added section to Supplemental Information,

“Kinetic isotope effect in abiotic ethane

Kinetic isotope effect may also play a role in ^{13}C - ^{13}C signature from abiotic samples associated with abiotic polymerization step. To estimate the kinetic isotope effect, three simplified polymerization reaction were considered as follows:

Reaction rate for each polymerization reactions can be defined using reaction rate constants k , k' , and k'' as follows:

$$d[^{12}\text{CH}_3^{12}\text{CH}_3]/dt = k[^{12}\text{CH}_3][^{12}\text{CH}_3] \quad (\text{S4})$$

$$d[^{12}\text{CH}_3^{13}\text{CH}_3]/dt = 2 \times k'[^{12}\text{CH}_3][^{13}\text{CH}_3] \quad (\text{S5})$$

$$d[^{13}\text{CH}_3^{13}\text{CH}_3]/dt = k''[^{13}\text{CH}_3][^{13}\text{CH}_3] \quad (\text{S6})$$

Then, $^{13}\text{R}_{\text{Ethane}}$ and $^{1313}\text{R}_{\text{Ethane}}$ can be obtained by dividing eq.S5 by eq.S4 and 2 and eq.S6 by eq.S4 considering the symmetry of two carbon atoms in ethane as follows:

$$^{13}\text{R}_{\text{Ethane}} = (k'/k) \times ^{13}\text{R}_{\text{Methyl}} \quad (\text{S7})$$

$$^{1313}\text{R}_{\text{Ethane}} = (k''/k) \times ^{13}\text{R}_{\text{Methyl}}^2 \quad (\text{S8})$$

where $^{13}\text{R}_{\text{Methyl}}$ represents $^{13}\text{C}/^{12}\text{C}$ ratio calculated from $[^{13}\text{CH}_3]/[^{12}\text{CH}_3]$. Consequently, the $\text{D}^{13}\text{C}^{13}\text{C}$ value can be calculated as follow:

$$\Delta^{13}\text{C}^{13}\text{C} = \ln[(k''/k)/(k'/k)^2] \quad (\text{S9})$$

Therefore, the $\Delta^{13}\text{C}^{13}\text{C}$ value associated with kinetic isotope effect through polymerization reaction depends on the coefficient $(k''/k)/(k'/k)^2$, leading to depleted ^{13}C - ^{13}C signature $[(k''/k) < (k'/k)^2]$ or enriched ^{13}C - ^{13}C signature $[(k''/k) > (k'/k)^2]$ relative to stochastic distribution. If $\Delta^{13}\text{C}^{13}\text{C}$ value has stochastic distribution, (k''/k) is equal to $(k'/k)^2$.”

L225: Clarify this or add more citations. What would cause ‘variable’ reversibility in these reactions? Under what circumstances would you expect them to be more or less reversible? If the reversibility is variable, its not clear that repeated cleavage and formation of the bonds would trend towards an equilibrium composition. Please expand on this prediction to further justify it, either here or in the supplement.

We have modified the sentence as follow (L251-L257):

Original text,

“It can be predicted that repeated cleavage and formation of C-C bonds in hydrocarbons leads to an isotopic exchange, where $\Delta^{13}\text{C}^{13}\text{C}$ of abiotic ethane shifts toward the homogeneous isotopic equilibrium (+0.22‰ at 25°C; Extended Data Fig. 2) (Fig. 2b).”

↓

Revised text,

“Fully reversible reactions may yield equilibrium isotope composition (the curved black line in Fig. 2b), whereas irreversible reactions tend to be governed by kinetic isotope effect as represented in the ethane synthesized by UV experiment (Fig. 2b). We suggest that cleavage of C-C bonds in hydrocarbons may

enhance the reversibility and leads to an isotopic exchange, where $\Delta^{13}\text{C}^{13}\text{C}$ of abiotic ethane shifts toward the homogeneous isotopic equilibrium ($\Delta^{13}\text{C}^{13}\text{C} = +0.22\text{‰}$ at 25°C ; Extended Data Fig. 2) (Fig. 2b).”

L231: The ‘increase in D13C13C’ referred to here is not clear. Increase as a function of what other variable? I’m assuming it’s the slope of the CSIA trend, but this is not clear here.

We have modified the sentence as follow (L260):

Original sentence,

“In summary, the observed $\Delta^{13}\text{C}^{13}\text{C}$ variations (as an increase in $\Delta^{13}\text{C}^{13}\text{C}$) in abiotic ethane...”

↓

Revised sentence,

“In summary, the observed low $\Delta^{13}\text{C}^{13}\text{C}$ values in abiotic ethane...”

L238: How long would the proposed isotope exchange take? Has it been modelled? If so, is the timescale reasonable for these gases? Recent research on methane suggests that hydrogen isotope exchange can take a very long time (at least at low temperatures in the absence of catalysts) (Turner et al, 2022, GCA). What is the the estimated age of production of these gases?

Unchanged. We thank the reviewer for pointing the issue. At present, we have no experimental data which $\Delta^{13}\text{C}^{13}\text{C}$ could be equilibrated completely under natural sites where abiotic synthesis occur and how long to take it. This might be important to ability of distinguish ethane between thermogenic and abiotic ethane because if ethane is completely equilibrated $\Delta^{13}\text{C}^{13}\text{C}$ will be overlapped. We are planning to run experiments starting from methane with catalysts to take advantage of the re-equilibration. However, this is beyond the scope of this paper.

L247: Mars is always an exciting target, but I’m not sure this method is feasible for application to Mars, so it seems kind of odd to discuss it here. Presumably this measurement isn’t going to happen with by Rover, so would it entail a gas sample return mission? Has anyone proposed this? I’d say if it isn’t realistic in the next few decades it probably shouldn’t be discussed as an implication of the research.

Remain unchanged. The ^{13}C - ^{13}C abundance might be used to distinguish biotic and abiotic processes not only in ethane gas but also potentially a variety of organic molecules containing C-C bonds. This paper represents a first insights into ^{13}C - ^{13}C clumping in organic molecules. This will help driving research to more complex organic such as lipids or amino acids. Therefore, we expect the approach to be applicable to sedimentary organic matter on Earth, but also on Mars in the case of a return mission is planned, or directly from a martian meteorite such as ALH 84001.

Fig 2. What does it mean that the natural gas data are aligned perpendicular to the temperature curve? Is there another process that could induce variation other than the temperature of formation? I think this deserves some discussion.

Good point. We have added the discussion as follow (L150-L155):

“The thermogenic ethane seems aligned perpendicular to the equilibrium temperature curve (Fig. 2a), which may potentially reflect the variation of $\Delta^{13}\text{C}^{13}\text{C}$ of organic precursor, though, at present, the available $\Delta^{13}\text{C}^{13}\text{C}$ data of organic precursor is only limited to bio-ethanol due to the analytical difficulties. Future studies should pursue $\Delta^{13}\text{C}^{13}\text{C}$ of organic molecules such as n-alkanes, fatty acids, and lignin to evaluate the $\Delta^{13}\text{C}^{13}\text{C}$ variations in organic precursor.”

L444: How is this lambda value known? Maybe a citation is necessary here, or more elaboration.

We have cited the reference (Taguchi et al., 2021) for clarity (L492).

L447: I think I understand this sentence, but it is written awkwardly. I think it's saying that we can only be confident in relative values between samples, not absolute values. Please re-word.

This comment was discussed in detail in the responses to the previous comments (Reviewer 1). We have deleted the sentence in the revised manuscript (L489-L493).

L612: I think it would be good to provide more details on previous evidence of ethane oxidation in these samples. Is there independent evidence indicating this process at this site? Based on other studies of ethane oxidation, is this a likely location? It would be nice to see independent corroboration.

Unchanged. Finding evidence for hydrocarbons oxidation in the subsurface is a difficult task. In addition, microorganisms oxidizing C₂₊ hydrocarbons anaerobically have been isolated only very recently (e.g. Chen et al, 2019, Nature, for ethane oxidation). Therefore, we rely on the position-specific isotope composition of propane shown in Gilbert et al. (2019) to be a reliable indicator of the anaerobic oxidation of C₂₊ hydrocarbons. Indeed, we believe that the new indicator presented in our paper may be useful to detect and even quantify the anaerobic oxidation of ethane.

Reference cited

Turner, Andrew C., et al. "Experimental determination of hydrogen isotope exchange rates between methane and water under hydrothermal conditions." *Geochimica et Cosmochimica Acta* (2022).

Reviewer #3 (Remarks to the Author):

This manuscript reports ethane clumped isotope data that show promising patterns for distinguishing abiotic and biotic ethane. The type of data presented here are of high interests to people in the field, and the implications should be quite impactful to the general readership of Nature Communications. I have a few suggestions that are mostly minor, except for some concerns about the standardization process. Pending these revisions, I would recommend publication.

We are grateful to the reviewer's enthusiastic comments.

L52: It is too parsimonious to attribute thermogenic ethane's near-stochastic clumping to biological precursors. In thermogenic ethane production, clumped isotope signatures could be a result of many factors: 1) KIE and statistical effects in ethane generation 2) precursor isotope distribution 3) KIE and statistical effects in ethane destruction (Peterson et al. 2018). 4) potential equilibration at higher maturities. The near stochastic distribution could be a result small effects in all of these categories, or nuanced cancellation these effects.

We have changed the sentence in response to the comment from the reviewer #2. We have not intended to say thermogenic ethane is near stochastic. Instead, biological precursor C-C could be near stochastic. For avoiding the confusion, we have added "presumably" at L51.

L80-81: More durable is too general. In thermal cracking, C—C bonds are much easier to break than C—H bonds. Please state what types of chemistry C—C bonds are more resistant to specifically.

Agreed. Following the reviewer's comment, we have changed the sentence to clarify as follow (L79-L81):

Original text,

“More robust information may come from ^{13}C - ^{13}C clumping in organic molecules, because C-C bonds are more durable than C-H bonds in hydrocarbons.”

↓

Revised text,

“More robust information may come from ^{13}C - ^{13}C clumping in organic molecules, because carbon in ethane is more durable than hydrogen, which is exchanged with surrounding water²¹.”

L86: “and compared them with”

Done.

L97: The methods of lab synthesis of abiotic gas do not include FTT. This is a little regretful, because FTT is more relevant for abiogenesis on Earth environments. I suggest that the authors give a brief overview of different abiotic synthesis in the introduction and describe the scope of this study, because the introduction only talks about FTT. This is also a big distinction that the broad science readership of nature communications might be unfamiliar with.

Unfortunately, we are not equipped to conduct FTT experiments, but an ongoing collaboration may extend our knowledge. Accordingly, we have modified the introduction to make it more general, as follows (L63-L66):

Original text,

“.....a variety of reactions (including the Sabatier, and Fischer–Tropsch-type reactions²⁻⁴) in both deep crustal fluids,”

↓

Revised text,

“.....a variety of reactions (including free-radical reactions, the Sabatier, and Fischer–Tropsch-type reactions^{2-4,18}) in both deep crustal fluids,”

Also, we have added a reference and the sentence to give the relevance of abiotic synthesis conducted in this study with natural sites we measured as follow:

L267-L269: “The similarity of ethane from the gamma radiolysis experiments to the Kidd Creek samples is notable given the proposed role of radiolysis in producing acetate and formate at that site (Sherwood Lollar et al., 2021).”

Added reference,

43. Sherwood Lollar, B. et al. A window into the abiotic carbon cycle - Acetate and formate in fracture waters in 2.7 billion year-old host rocks of the Canadian Shield. *Geochim. Cosmochim. Acta* **294**, 295–314 (2021). <https://doi.org/10.1016/j.gca.2020.11.026>.

L98: “can be” should be “are”

Done.

L127-128: These parameters can be moved into the Methods.

Unchanged. We believe that these parameters are important to follow the reasoning here.

L164–165: The highest $\Delta^{13}\text{C}_{13}\text{C}$ sample from Tokamachi corresponds to an equilibrium temperature well below 0C, making equilibration unlikely.

Yes. This discrepancy could be derived from the uncertainties of assumption of stochastic reference frame, temperature of microbial ethane oxidation and degree of reversibility in natural populations. Those factor except for the reference frame are discussed in our manuscript.

We have changed the sentence in response to the comment from the reviewer #1 and have added the sentence to clarity (L181-L182).

L182-L183: “based on the assumption of stochastic reference frame (see Method).”

L188–191: That’s right. It is hard to explain why the UV experiments have $\Delta^{13}\text{C}_{13}\text{C}$ around -0.9 per mil, while the prediction of this collision frequency theory is at -0.52 per mil. Although it could possibly arise from the error in absolute reference frame.

Yes. We have added the sentence as follow:

L201: “In a stochastic reference frame assumed here (see Method)”

L207: If you add calculation of the collision KIE with the third body M, would the clumped isotope effect shift much?

Good point. We have added the section to Supplemental Information in response to the comment from the reviewer #1 as follow:

Added section to Supplemental Information,

“The recombination reaction between CH_3 radical is via the formation of C_2H_6^* with high energy (S10) and deactivation to C_2H_6 by its collision with another molecule/atom (third body) M (S11) as follow:

The pressure of our experiments (about 12kPa of CH_4) is range of high-pressure-limit, leading to an almost quantitative conversion of C_2H_6^* to C_2H_6 . Therefore, the third body, M, does not contribute to the isotope effect during the experiment here.”

L244–247: Titan is a more relevant place for the photolysis experiments studied in this work. Titan’s organic synthesis also starts from methane like in this study.

Agreed. We have changed the sentence and added reference as follow:

Original text,

“...such as Mars”

Revised text,

“...such as Mars, Titan, and Enceladaus^{7,16,44}”

Added reference,

44. Lunine, J., Stevenson, D. & Yung, Y. Ethane ocean on Titan. *Science*. 222, 1229–1230 (1983). <http://doi.org/10.1126/science.222.4629.1229>.

L462–465: “the aldolization and ketolisation reactions are partially at equilibrium(44).” and “These C-C bonds are produced in an aldolisation reaction, which is thought to be at equilibrium.” These two sentences are conflicting as the first one says partial equilibrium but second says full equilibrium. Besides, It is unclear to me how C-5 and C-6 positions are controlled by the reversible reaction of aldolase, because Gilbert et al. (2013) only applied equilibrium analysis to C-3 and C-4 positions. Please elaborate. Another caveat of this treatment is that there might be KIE in ethanol fermentation, unless the authors could prove that the reaction is non-fractionating or quantitative.

We thank the reviewer for pointing us of two contradictory sentences. Mondal et al., 2015 (our reference in number 44) proposed the aldolisation and ketolisation reactions which make the C1-C2 and C5-C6 positions of glucose are potentially at equilibrium. We have removed ambiguous wording and changed the sentence as follow to clarity (L507-L511):

Original text,

“...the aldolisation and ketolisation reactions are partially at equilibrium⁴⁴. The measured ethanol²² is derived from the carbons in the C1-C2 and C5-C6 positions of glucose. These C-C bonds are produced in an aldolisation reaction, which is thought to be at equilibrium⁴⁴.”

Revised text,

“...the aldolisation and ketolisation reactions are at equilibrium⁴⁴. The measured ethanol²² is derived from the carbons in the C1-C2 and C5-C6 positions of glucose which are produced in an aldolisation reaction⁴⁹.”

In addition, Gilbert et al., 2011 and Bayle et al., 2015 have shown that the isotope fractionation from glucose to ethanol is negligible which is discussed in our previous study in Taguchi et al., 2020.

L469: The difficulty to perform DFT calculations on glucose is not so much higher than doing so on C2 molecules...

It is rather a matter of server time than difficulty *per se*. The theoretical calculation has been done by one of our co-authors, Qi Liu. The time we can spend on these calculations is limited and therefore we decided to use an analog molecule. Ideally, these calculations would be made not in the gas phase, but with surrounded water molecules. But again, this will necessitate a long calculation time.

L810–811: Just say “eq. 9–10”

Done.

Original text,

“The $\Delta^{13}\text{C}^{13}\text{C}$ values are calculated from equilibrium constants between singly substituted species, a doubly substituted species, and an un-substituted species (see Methods).”

↓

Revised text,

“The $\Delta^{13}\text{C}^{13}\text{C}$ values are calculated from equilibrium constants from eq. (9) and eq. (10) (see Methods).”

REVIEWERS' COMMENTS:

Reviewer #1 (Remarks to the Author):

I've had a chance to go through the authors' revisions, and I believe they have made good-faith efforts to address all of my comments. Personally, I would have conducted a couple more experiments and measurements described in my review before I published this paper, but that is a judgement call and I'm happy to recommend the revised work be published as is.

Reviewer #2 (Remarks to the Author):

The authors have addressed my main critiques and I think the article is nearly ready for publication. However, there are a few, mostly minor, changes I would like to see in the final version. Mostly these are about enhancing clarity of the language. In one case there are more substantive issues raised by another reviewer that I think need to be addressed.

L45: "rely on the independence from the stable isotopic composition of the starting material". This language is confusing. I assume the authors mean something like 'because it is independent of the isotopic composition of the starting material'.

L80: I don't think 'durable' is the right word here. Maybe 'less readily exchanged'.

L104: This paragraph is important, and I don't think the differences between methods are adequately addressed. Ultimately, we can only rely on these data if they are reproducible, and this analysis of the pyrolysis results raises major questions about that. I don't think the authors need to fully address the question of reproducibility here. However, to simply say that the reasons for the discrepancy are unknown is too vague and simplistic. I think at the least the authors should provide testable hypotheses to explain why there is this difference, which would provide a template for future work. Some key questions: are the experimental conditions identical, or are there key differences that could explain the different results? Why is $d^{13}C$ changing by so much while the clumped isotope values remain the same, and is this similar in the results of Clog et al? Can the authors propose theoretical reasons why their method would produce such different results? Is it possible this is a result of the combinatorial effects related to flourination discussed by reviewer 1?

Overall, I think some answers to these questions are needed in order to be confident in these results and to move forward in addressing this discrepancy.

L146: "Narrow range" is too vague. Please provide a quantitative range. Readers shouldn't have to go back to the previous paper to look it up, since this is an important point.

L153: I think 'precursor' needs to be changed to 'precursors' to make this sentence grammatically correct. Or if it is singular, it should be 'the precursor'. Analytical difficulties is vague and imprecise; please indicate more specifically why it is not possible to analyze these other biomolecules currently.

L263: This sentence is not grammatically correct. I think the word 'and' should be removed.

Reviewer #3 (Remarks to the Author):

The authors have improved their manuscript addressed my concerns brought up in the initial review. I recommend publication.

Comment and Reply

REVIEWER COMMENTS

Reviewer #1 (Remarks to the Author):

I've had a chance to go through the authors' revisions, and I believe they have made good-faith efforts to address all of my comments. Personally, I would have conducted a couple more experiments and measurements described in my review before I published this paper, but that is a judgement call and I'm happy to recommend the revised work be published as is.

Reviewer #2 (Remarks to the Author):

The authors have addressed my main critiques and I think the article is nearly ready for publication. However, there are a few, mostly minor, changes I would like to see in the final version. Mostly these are about enhancing clarity of the language. In one case there are more substantive issues raised by another reviewer that I think need to be addressed.

L45: "rely on the independence from the stable isotopic composition of the starting material". This language is confusing. I assume the authors mean something like 'because it is independent of the isotopic composition of the starting material'.

Done.

L80: I don't think 'durable' is the right word here. Maybe 'less readily exchanged'.

Done.

L104: This paragraph is important, and I don't think the differences between methods are adequately addressed. Ultimately, we can only rely on these data if they are reproducible, and this analysis of the pyrolysis results raises major questions about that. I don't think the authors need to fully address the question of reproducibility here. However, to simply say that the reasons for the discrepancy are unknown is too vague and simplistic. I think at the least the authors should provide testable hypotheses to explain why there is this difference, which would provide a template for future work. Some key questions: are the experimental conditions identical, or are there key differences that could explain the different results? Why is $\delta^{13}\text{C}$ changing by so much while the clumped isotope values remain the same, and is this similar in the results of Clog et al? Can the authors propose theoretical reasons why their method would produce such different results? Is it possible this is a result of the combinatorial effects related to fluorination discussed by reviewer 1?

We agree with the reviewer's suggestion that we should explain the discrepancy more in details. The results from the pyrolysis experiments at the same 600°C temperature using the same apparatus (quartz reaction vessel) give different changes in \$\Delta^{13}\text{C}^{13}\text{C}\$ values but with a similar \$\delta^{13}\text{C}\$ shift. This suggests a discrepancy between the two methods for isotopologues analysis (Fluorination vs HR-IRMS). One possibility is that the \$\Delta^{13}\text{C}^{13}\text{C}\$ scales could be different for the two methods. In our case,

potential scale compression has already been rigorously discarded in a previous paper using spiked samples (Taguchi et al., 2021). In any case, further interlaboratory comparisons will be necessary to address the question of reproducibility. These points are added in the revised manuscript:

Original paragraph,

“A previous study using high-resolution mass spectrometry exhibited 4‰ variation of $\Delta^{13}\text{C}^{13}\text{C}$ values of thermogenic ethane²⁵. This variation has been suggested to arise from the pyrolysis of ethane, which leads to a decrease in $\Delta^{13}\text{C}^{13}\text{C}$ values^{25,26}. The data obtained by our method show a narrower range of the $\Delta^{13}\text{C}^{13}\text{C}$ values (0.57‰). A pyrolysis experiment conducted in the same conditions showed no change in the $\Delta^{13}\text{C}^{13}\text{C}$ values (see SI). These observations point to a discrepancy between the two methods, the reason for which is currently unknown and will necessitate further interlaboratory comparisons. Presently, the comparison of data between the two methods would thus be inaccurate. Hence, while we consider the calibration of both methods an important problem to solve in the future, the data presented here will be that obtained by the method presented in ref²².”

↓

Revised paragraph,

“A previous study using high-resolution mass spectrometry exhibited 4‰ variation of $\Delta^{13}\text{C}^{13}\text{C}$ values of thermogenic ethane²². This variation has been suggested to arise from the pyrolysis of ethane, which leads to a decrease in $\Delta^{13}\text{C}^{13}\text{C}$ values^{22,23}. The present study using a conventional isotope ratio mass spectrometry after conversion of C_2H_6 to C_2F_6 shows a narrower range of the $\Delta^{13}\text{C}^{13}\text{C}$ values (0.57‰). Our pyrolysis experiment conducted at the same temperature as in ref²² (600°C) and using a similar quartz vessel showed no change in the $\Delta^{13}\text{C}^{13}\text{C}$ values in contrast to ref²² (Supplementary Fig. 1, Supplementary Table 3). These observations point to a potential discrepancy between the two methods for isotopologues analysis. Further interlaboratory comparisons will be necessary to calibrate the data from the two methods. The data presented here will be obtained solely by the method presented in ref¹⁹ which gives reproducible $\Delta^{13}\text{C}^{13}\text{C}$ values with no scale compression²⁰ (see Methods).”

”

Overall, I think some answers to these questions are needed in order to be confident in these results and to move forward in addressing this discrepancy.

L146: "Narrow range" is too vague. Please provide a quantitative range. Readers shouldn't have to go back to the previous paper to look it up, since this is an important point.

Done.

L153: I think 'precursor' needs to be changed to 'precursors' to make this sentence grammatically correct. Or if it is singular, it should be 'the precursor'. Analytical difficulties is vague and imprecise; please indicate more specifically why it is not possible to analyze these other biomolecules currently.

Done.

L263: This sentence is not grammatically correct. I think the word 'and' should be removed.

Done.

Reviewer #3 (Remarks to the Author):

The authors have improved their manuscript addressed my concerns brought up in the initial review. I recommend publication.